# Egalitarian preferences in young children depend on the genders of the interacting partners
Marijn van Wingerden [1,2,5] ✉, Lina Oberließen[3,4,5] & Tobias Kalenscher [3]

In decisions between equal and unequal resource distributions, women are often believed to be more prosocial than men. Previous research showed that fairness attitudes develop in childhood, but their— possibly gendered, developmental trajectory remains unclear. We hypothesised that gender-related fairness attitudes might depend not only on the gender of the Allocator, but also on that of the Recipient. To examine this, we tested 332 three to 8-year-old children in a paired resource allocation task, with both boys and girls acting as Allocators and Recipients. We indeed found gender-related effects: girls more than boys aimed to reduce advantageous inequity, and Allocators of both genders were more averse against male Recipients being better off. Notably, older girls exhibited an envy bias, i.e., they tolerated disadvantageous inequity more when the resource allocation was in favour of other girls than when it favoured boys. We also observed a gender-related spite gap in boys aged 7-8: unlike girls, boys treated other boys with spite, i.e., they valued unfair distributions in their own favour over equal outcomes, especially if rejecting advantageous inequity was costly. This pattern hints at contextualised gender-related fairness preferences that evolve with age that could depend on same- and cross-gender past interaction experiences.

A mechanism for detecting unfairness is essential for establishing and maintaining long-term cooperation in larger groups[1]. Humans and several other social species reject disadvantageous unequal reward distributions (disadvantageous or 'first-order inequity aversion (IA)': the dislike of being worse off than others, given comparable efforts[2–4]). But only humans and, possibly, very few other species have developed a complete fairness concept, including the rejection of unequal advantageous distributions (advantageous or 'second-order IA': the dislike of others being worse off[2,5–8] but see refs. 9,10). In humans, IA is often studied through resource allocation tasks in which predefined fair or unfair resource allocations can be either accepted or rejected[11–17]. Although not every study on IA tests or reports gender-related differences, many investigations reveal that adult women often exhibit more compassionate behaviour in decision-making, aimed to minimise inequity between members of interacting dyads. In contrast, men show more competitive behaviour, which could be labelled (on a surface level) as envy or even spite, i.e., they tend to maximise own gains, or even accept costs to minimise disadvantageous inequity[18–26], but see ref. 27. In line with the Sex and Gender Equity in Research (SAGER) guidelines for the use

of the terms *sex* and *gender*[28], here, we consistently use the term *gender* (limited to the binary terms 'girl' and 'boy' for individuals of female/male sex) instead of *sex* in our description of previous research and interpretations of our own results to reflect the fact that fairness preferences are most likely shaped by a combination of socio-economic, cultural, experiential and genetic factors. However, most studies, including ours, rely on biological sex as a proxy for (binary) gender and it remains an open question how fairness preferences relate to a (multidimensional) gender spectrum[29].

Recent insights suggest that such adult gender-related differences in social preferences are acquired during childhood and adolescence[21,30] but see ref. 31. Children start to show egalitarian preferences (that is, prefer equal over unequal outcomes) between the ages of 3 and 8[13–16,32–35], a period that coincides with differentiation in gender-specific behaviours[36–38], but when and how fairness preferences start to diverge between genders is still an open question that is topic of current research agendas[30,39]. Recent theories posit that social preferences such as social norms, are likely shaped, among others, by past social interaction experiences between boys and girls (as far as we are aware, non-binary identifying children are not included in these theories)

[1]Social Rodent Lab, Institute of Experimental Psychology, Heinrich-Heine-University Düsseldorf, Düsseldorf, Germany. [2]Cognitive Science & Artificial Intelligence Department, School for Humanities and Digital Sciences, Tilburg University, Tilburg, The Netherlands. [3]Comparative Psychology, Institute of Experimental Psychology, Heinrich-Heine-University Düsseldorf, Düsseldorf, Germany. [4]University of Veterinary Medicine, Vienna, Austria. [5]These authors contributed equally: Marijn van Wingerden, Lina Oberließen. ✉e-mail: E.J.M.vanWingerden@tilburguniversity.edu

during development[21,40]. When interaction partners reveal their fairness preferences, children can learn about their egalitarian preferences from direct experience that might shape and possibly gender their own egalitarian attitudes through generalisation. Most studies we reviewed used preference elicitation methods that omitted information about the interaction partners' genders, as for example in hypothetical or anonymous partner scenarios[14,33,34,41–56] and only few studies controlled the gender of *all* interaction partners, usually in a gender-matched pairing design[16,57–59]. It thus is an open question whether differences in fairness preferences in resource allocation tasks depend on the genders of all interaction partners. Hence, to date, the effect of the genders of both interacting partners on egalitarian preferences during development remains elusive, even though it is known from other experimental contexts, e.g. bargaining, that the exact configuration of genders to the roles in the economic context, more than gender per se, plays an important role in explaining the observed behaviour[60]. It thus might well be possible that egalitarian preferences of children also differ with respect to their own gender *and* the gender of their interaction partner.

To better understand how the genders of the interaction partners influence choices, thought to reflect egalitarian preferences during social interaction, we explored the development of these preferences during childhood. We used an established resource allocation task that is widely used to measure social preferences in children[13,17,61]. An Allocator was paired with a known Recipient, and in four dilemmas that forced participants to reveal their fairness preferences, the Allocator decided between costly or non-costly equal outcomes vs. advantageous or non-advantageous unequal reward distributions (see below).

To disentangle the effects of the gender of the Allocator and that of the Recipient on fairness preferences in these dilemmas, we employed a $2 \times 2$ design where, across pairings, children of both genders were assigned to either the Allocator or the Recipient role (but roles did not change for any particular child). Note, though, that we did not poll the participants on their socially constructed (continuous) gender identity but categorised them solely based on their (assumed) binary biological sex. Data are thus presented using biological terms (female/male participants), and are described disaggregated for all combinations of female/male participants[62], but the interpretation of our results is phrased in terms of gender-related differences between girls and boys.

When analysing these choices, we focused on Age effects (both as a continuous variable and in 3 subsamples: 3–4 years, 5–6 years and 7–8 years) and on explicitly splitting the data according to the gender of the Allocator and Recipient (i.e., 4 subsamples: female-female, female-male, male-female and male-male dyads). To study the development of gendered fairness preferences over age groups, the full sample was thus split in 12 subsamples total (for the group-based analysis), or in 4 Dyad-gender groups (when analysing Age as a continuous variable). Based on previous research, we aimed to replicate the effect of Age on fairness preferences and furthermore hypothesised that the assumed development of gendered fairness preferences would show up as clear differences in fairness preferences that depended both on the gender of the Allocator and the Recipient.

## Methods
### Participants
We tested 332 children between 3 and 8 years (females = 176, males = 156; mean age = 71.95 months, *s.e.m.* = 1.05 months, range = 37 – 111 months). Data from 1 child was lost, thirty-two children who could not answer all comprehension questions correctly (see below) and 20 children who had a distinctly positive ($N = 6$ pairs, 12 children) or negative relationship ($N = 4$ pairs, 8 children) with their assigned partner were excluded from data analysis, resulting in a final sample of $N = 279$ Allocator/Recipient pairs. When analysing the distribution of choices (accept/reject unequal distribution) per dilemma (1 choice per child, for each of the 4 dilemma's) across children, we did not find a significant difference in distributions between the full sample and the final included sample (Equality of proportions test; DI Non-Costly $\chi^2_1 = 0.780$, $p = 0.377$; DI Costly $\chi^2_1 = 0.609$, $p = 0.435$; AI Non-Costly $\chi^2_1 = 0.003$, $p = 0.9556$; AI Costly $\chi^2_1 = 0.000$, $p = 1$[28,62], see Table 1).

The remaining sample of 279 children (females = 146, males = 133) was separated into three age groups: (1) 3–4 years old (39–59 months): females = 32, males = 33, mean age = 51.46 months, *s.e.m.* = 0.68; (2) 5–6 years old (60–83 months): females = 56, males = 56, mean age = 72.60 months, *s.e.m.* = 0.70; (3) 7–8 years old (84–111 months): females = 58, males = 44, mean age = 93.11 months, *s.e.m.* = 0.62. The relatively lower sample size of the youngest age group results from a higher exclusion rate due to comprehension problems (inability to answer all comprehension questions correctly; see below).

Data were collected in five primary schools and eight daycare facilities for children in urban, middle- to upper-middle class areas (Düsseldorf, Germany). With the consent of the school/daycare facility administration, information letters were sent to the parents of the children requesting permission for their child´s participation in the study. In these, the parents were informed about the experimental procedure, anonymization, and data storage policies. We only included children whose parents had given written consent to participate in our study. Our study was approved by the Ethics Committee for non-invasive human research of Heinrich-Heine-University, Düsseldorf. Our study was not preregistered.

**Table 1 | Sample size and proportions of subgroups in the dataset; F: female, M: male**

| Age | N | % | Gender Allocator | N | % | Dyad | N | % in Dyad | % Total |
|---|---|---|---|---|---|---|---|---|---|
| 3–4 year | 65 | 23.3% | F | 32 | 49.2% | F-> F | 15 | 23.1% | 5.4% |
| | | | | | | F-> M | 17 | 26.2% | 6.1% |
| | | | M | 33 | 50.8% | M-> F | 14 | 21.5% | 5.0% |
| | | | | | | M-> M | 19 | 29.2% | 6.8% |
| 5–6 year | 112 | 40.1% | F | 56 | 50.0% | F-> F | 34 | 30.4% | 12.2% |
| | | | | | | F-> M | 22 | 19.6% | 7.9% |
| | | | M | 56 | 50.0% | M-> F | 28 | 25.0% | 10.0% |
| | | | | | | M-> M | 28 | 25.0% | 10.0% |
| 7–8 year | 102 | 36.6% | F | 58 | 56.9% | F-> F | 27 | 26.5% | 9.7% |
| | | | | | | F-> M | 31 | 30.4% | 11.1% |
| | | | M | 44 | 43.1% | M-> F | 27 | 26.5% | 9.7% |
| | | | | | | M-> M | 17 | 16.7% | 6.1% |
| TOTAL | 279 | | | | | | | | 100% |

**A**

**B**

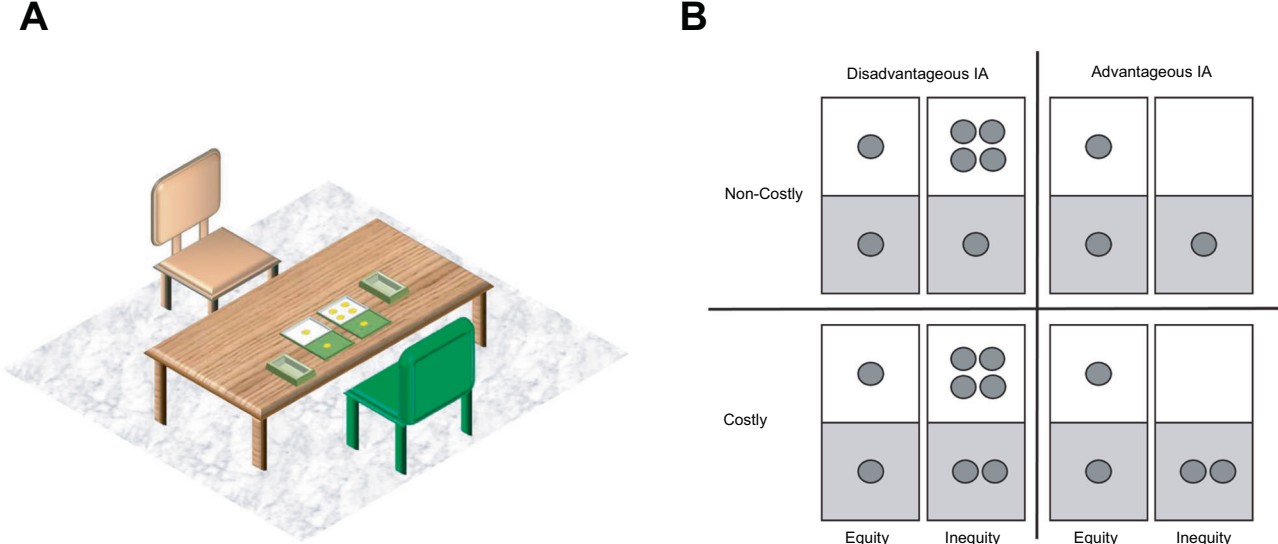

**Fig. 1 | Experimental arrangement of the inequity aversion choice task. A** The experimenter (brown chair) sat opposite the subject (Allocator child, green chair) performing the IA choice task while the Recipient child worked on the distraction task at the same time in the same room. **B** Reward distributions differed in type of inequity and cost. In each trial, the Allocator child selected one of two boxes with different reward distributions. The grey part of each box (green in **A**) depicts the Allocator´s outcome, the white part the Recipient´s outcome. All unfair choice options were pitted against a fair 1:1 alternative. The unfair distributions yielded either disadvantageous IA (left quadrants), or advantageous IA (right quadrants) outcomes and were either non-costly (top quadrants), or costly (bottom quadrants).

## Procedure and apparatus

Children participated in the study in same- or cross-sex pairs. Previous studies (reviewed in ref. 17) have used different criteria for pairing children and assigning roles in their experiments. For example, many studies use anonymous partners in dyads or triads (e.g.[33,34,41,42,44–48,63], and others). The use of anonymous partners is beneficial for several reasons, e.g. to minimise potential reputational concerns, such as merit considerations, effort and need that could influence children's decisions above and beyond IA[17,49,52–54,64]. However, the anonymity of the partner presupposes that the children possess a certain degree of cognitive abstraction ability. The ability to picture, and act on, anonymous partners may be differentially pronounced, particularly in younger children; it may, therefore, be a potential source of age bias. To circumvent these problems, other studies used siblings or friends as partners (e.g.[65]), with the obvious disadvantage that previous interaction experiences, reciprocation expectations and reputational concerns with known partners that have a relationship with the acting child are potential confounds of social preferences[17,66]. To avoid all these potential disadvantages while minimising the requirement for abstraction, we only considered pairs of real children that had little or no previous connection: none of them were friends or felt aversion against each other. We therefore opted for pairs from different groups/classes so that there was no relationship between children. However, this was not feasible in all facilities. For pairings from one and the same group we asked the responsible caretaker for an evaluation of the relationship of the paired children on a 10 cm-rating-scale from −5 to +5 afterwards (−5 = maximally negative relationship, 0 = neutral relationship, +5 = maximally positive relationship). We excluded children with distinctly positive or negative relationships represented by values above +2.5 or below -2.5.

The IA choice task took place on a table in a separate room within the particular facility. It consisted of two choice boxes with two equally sized compartments of different colours (white and green) and two separate collection boxes (Fig. 1A). Yellow smiley stickers were used as reinforcers to construct, in total, four reward distributions (Fig. 1B). We manipulated the type of inequity (advantageous versus disadvantageous) and cost (costly versus non-costly equal outcomes, relative to the own-outcome in the unequal distribution). Two unequal distributions present disadvantageous or advantageous inequity choices. Similarly, two distributions were non-

costly or costly and all choice options were pitted against a fair 1:1 alternative resulting in a $2 \times 2$ choice design (see Fig. 1). Thus, unlike the device introduced by Blake et al[14,16]., rejecting an unequal distribution in our experiments did not result in zero payoffs, but in the equal (1:1) alternative. That is, where *all* rejections in ref. 14,16 are costly, in our case, some rejections are non-costly, such as choosing the 1:1 alternative over the 1:4 disadvantageous distribution. Importantly, varying the costs of inequity rejection to the Allocator allowed us to model the subjective (dis)utility of inequity (see Fehr-Schmidt analyses below).

The whole experimental procedure followed a standardised protocol. The children were welcomed and asked if they wanted to participate. They were informed that the current study was a university project to investigate how children make decisions and distribute rewards (yellow smiley stickers) between themselves and another child by choosing one of two boxes with different distributions of stickers. It was randomly decided which child started with the IA choice task. The experimenter was always the same female (cis-gendered) person. She sat opposite the subject in the IA choice task and first informed the participants that they could stop the experiment any time. She explained that in each box, one side (white) contains the stickers for the other child (Recipient´s name is used), whereas the other side (green) contains the stickers for the Allocator (the child making the decision). The number of trials was not communicated but children were informed that they could keep the stickers subsequent to the experiment. For all four trials, the experimenter verbally informed the participant of the number of stickers for each child in each box. Before children made their decision by pointing at one of the boxes, they had to repeat the number of stickers they themselves and the other child would receive in each option. This comprehension question allowed us to evaluate whether children understood the task. After each choice, the experimenter transferred the stickers from the selected decision box to the collection boxes without any feedback and arranged the next distribution in the choice boxes. The order of distributions as well as the presentation side (left or right) of the equal distribution was counterbalanced among children. After the last decision of the first child in its role as Allocator, the stickers from the collection boxes were put in envelopes. Children switched position and the second child likewise performed the decision task. Envelopes were handed over to the subjects after the second child had also finished the decision task and all

stickers were collected in the envelopes. Depending on their choices, the number of stickers per child varied between 6 and 16. The effort to make decisions and perform the task was the same for all task conditions and participants.

The picture distraction task (German version of 'Where´s Waldo', Martin Handford, 8th edition, 2010) took place in the same room at another table, or on the carpet on the floor. However, tables were arranged in a way that the children were not sitting within their field of view to avoid any interaction and to keep choices private. The Recipient child worked on the picture distraction task together with a second experimenter, alone or with a local teacher. They were briefed to try to find Waldo on the pictures, highlight him with a marker and then turn over to the next page.

## Data analysis

Our initial analysis focused on comparisons of Generalised Linear Mixed-Effect Models (GLMMs; with logistic functions predicting the choice for the equal outcome as '1') with increasing complexity, using combinations of the following factors: Age (continuous), AllocatorGender (Female/Male) and RecipientGender (Female/Male) as between-subject factors and CostType (Costly/Non-costly) and InequityType (DI/AI) as within-subject factors. We used the glmer implementation from the lme4 package (v.1.1-29) with the bobyqa and Nelder_Mead optimisers. All reported models converged. As the GLMMs necessarily treat each choice as a binary outcome, no residual plots are included and this also precluded standard tests of IID distribution of residuals. However, linear-mixed models have been shown to be very robust against distributional assumption violations, where violations induce imprecision but not systematic bias in coefficients[67]. For all statistical tests, the level of significance was predefined as $p < 0.05$ if not otherwise specified. For multiple comparisons, $p$-levels were Bonferroni corrected.

We also employed a non-parametric, model-based approach using the Fehr-Schmidt model (Formula 1[5]), that yields two inequity aversion parameters from a set of decision problems that vary in own- and other-payoff. The idea is that the subjective utility of an outcome can be decreased both by being worse off than an interaction partner (disadvantageous inequity aversion (DI); weighted by the $\alpha$ parameter) *and/or* by being better off than an interaction partner (advantageous inequity aversion (AI); weighted by the $\beta$ parameter). Note that $\alpha$-parameter values (measuring DI) are sometimes labelled 'envy' and $\beta$-parameter values (measuring AI) are similarly often labelled 'compassion', or with negative sign, 'spite' (e.g.[5,13,33,35,68].). This model thus condenses all four choice problems into two parameters.

$$U_i(x) = x_i - \alpha_i \max\left\{x_j - x_i, 0\right\} - \beta_i \max\left\{x_i - x_j, 0\right\}, i \neq j \quad (1)$$

In the formula, $U_i(x)$ represents the utility of outcome x to Allocator i as a function of the magnitude of x, reduced by the number of units the allocator $i$ is worse off relative to the payoff to Recipient $j$ ($x_j - x_i$, set to 0 when difference is 0 or negative) weighed by the $\alpha$-parameter, and reduced by the amount of units the Allocator is better off than the Recipient ($x_i - x_j$, set to 0 when difference is 0 or negative) weighed by the $\beta$-parameter. One choice can thus only load on the $\alpha$- or $\beta$-parameter (depending on whether $x_j > x_i$ or $x_i > x_j$), and one needs a set of choices that features both disadvantageous and advantageous unequal options to concurrently estimate both parameters. The estimated weighing parameters $\alpha$ and $\beta$ capture the individual sensitivity to disadvantageous inequity ($\alpha$), or advantageous inequity ($\beta$), respectively, that is, how much the utility of x is reduced by each type of inequity. Note that we allow $\beta$-parameter values to become negative, indicating that individuals might derive utility from lowering the other player's payoff, even at a cost[33–35].

Because children made only one choice per outcome distribution, it is not possible to model $\alpha$ and $\beta$ at the individual subject level because softmax estimation (see Formula 2 below) would expect a probability for choosing the equal outcome, rather than a binary choice. However, for the entire study population or specific subgroups (e.g. 3–4-year-old girls), $\alpha$ and $\beta$ parameters can be estimated by averaging equity choices across individual

decisions in a given subgroup and treating this as the probability of choosing the equal outcome for that group. This yields a single best-fit parameter estimate per group. To subsequently also estimate the uncertainty in $\alpha$ and $\beta$ for a given subgroup, we applied a bootstrap approach with resampling, essentially repeating the modelling step for a randomised subsection of the original group and aggregating the obtained $\alpha$ and $\beta$ values in a distribution from which we report the mean and variance and thus construct confidence intervals on the parameters for inference between subgroups.

Thus, to obtain a distribution of choices within subgroups, we sampled $N = 5000$ draws of 150 randomly selected choices (with resampling) within the particular subgroup. For example, the target group of interest could be (1) all choices made by children in the middle age group, (2) all choices made by Allocators (male or female) paired with female Recipients, or (3) all choices made by Allocators in male-male dyads in the highest age group. For each bootstrap, the $N = 150$ choices were pooled and averaged. The resulting percentage choices for the equal alternatives were fit using a least-squares regression method optimising the parameters of a sigmoidal softmax decision function linking the utility differences through the noise parameter $\mu$:

$$P(Equity) = \frac{1}{1 + e^{\mu * (U_i - U_e)}} \quad (2)$$

where $U_i$ is the utility of the unequal option (see Eq. (1) above), $U_e$ is the utility of the equitable option (the 1/1 distribution) and $\mu$ is the noise parameter indicating choice inconsistencies (the lower $\mu$, the higher the inconsistencies).

Different to Fehr and Schmidt (1999), we did not place a limit on either $\alpha$ or $\beta$ and, as mentioned above, also allowed $\beta$ to be <0 to capture spite (negative compassion; i.e. they preferred unfair distributions in their own favour over equal outcomes) occasionally reported in children[33–35]. We rejected and re-drew bootstraps iterations where the fit returned a $\mu$ parameter smaller than 0.2, indicating very large differences between $U_i$ and $U_e$ due to extreme and unreasonable values in $\alpha$ and/or $\beta$. The resulting $\alpha$ and $\beta$ parameter bootstrap distributions followed a normal distribution. This allowed us to define a population mean and a standard deviation (to be construed as the bootstrapped standard error, shown as error bars in figures with a bootstrap distribution).

## Statistics and reproducibility

Test statistics, DF and exact $p$ values are reported for chi-square and likelihood ratio tests. Bootstrap distributions are compared to reference levels with one-samples Z-tests. Between-group differences are assessed with permutation tests with empirical $p$ values (e.g. percentile markers referenced to the permutation distribution). The resulting $p$ values are adjusted with Bonferroni correction for multiple testing.

For statistical comparisons of the differences between in Fehr-Schmidt model parameters estimated for the aggregated choices per subgroup (for example, the difference in Fehr-Schmidt parameters for the subgroup of male Allocators partnered with male Recipients vs. the subgroup paired with female Recipients), we added a permutation step in each bootstrap. Briefly, following this example, all choices involving male Allocators were extracted, and this sample split according to the gender of the Recipient. To assess what the range of possible parameter *differences* would be for these samples, a bootstrap procedure was used. In each bootstrap run, the binary labels indicating the target variable (Recipient gender in this example) were shuffled and reassigned in a randomly permuted way to the choices, thus keeping the number of choices accepting or rejecting inequity intact. Then, the sample was split according to the shuffled target variable and the Fehr-Schmidt model was run. Each bootstrap draw thus resulted in a paired, permuted parameter estimate for the two groups under comparison, from which the difference was retained. The resulting difference distribution indexed the range of putative differences for $\alpha$ and $\beta$ values, respectively, between the subgroups if allocation had been random. The shape of the distribution of possible differences in $\alpha$ and $\beta$ parameters followed a normal distribution and was expected (and found to be) centred on zero. The real

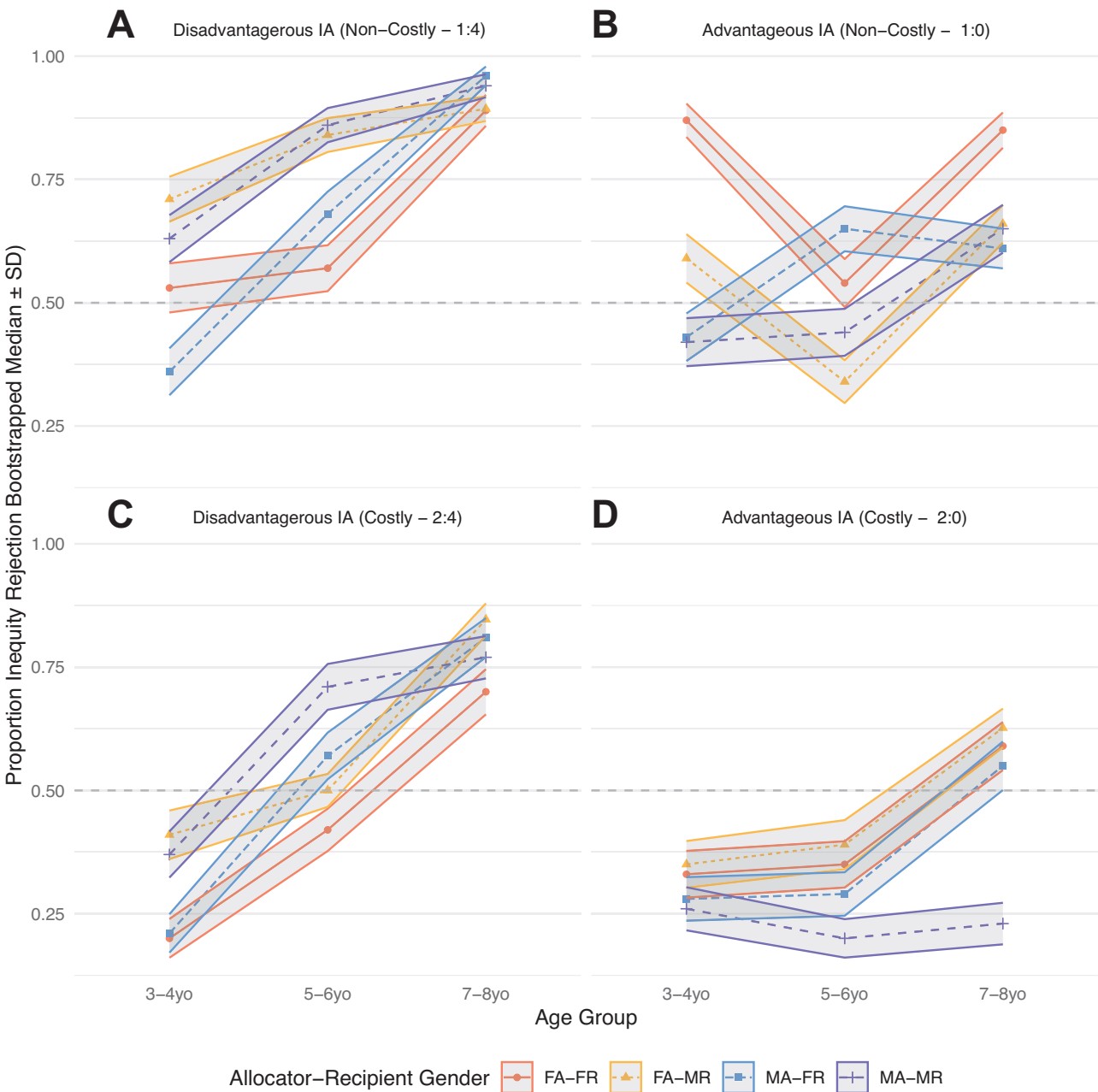

**Fig. 2 | Proportion of inequity rejections by choice type.** Figure shows the mean proportion of inequity rejections by choice type ($N$ = 4 dilemma's) and age group for each Dyad gender ($N$ = 4 combinations). Data points indicate the mean proportion of rejections per age group, shading indicates the bootstrapped standard errors (Standard Deviation of $N$ = 5000 draws of $N$ = 150 choices, with replacement).

Columns separate disadvantageous IA (DI, (**A** + **C**)) from advantageous IA (AI; (**B** + **D**)), rows separate Non-Costly (**A** + **B**) from Costly (**C** + **D**) dilemmas. Dashed horizontal lines indicate inequity indifference (50% choice of unequal and equal alternatives). FA Female Allocator, MA Male Allocator, FR Female Recipient, MR Male Recipient.

difference for both the $\alpha$ and the $\beta$ parameter between the subgroups of the original sample was also calculated and then compared to the reference permutation distribution of parameter differences for the calculation of empirical significance levels, using two-tailed confidence intervals for hypothesis-free comparisons, and one-tailed confidence intervals for directed hypotheses based on previous results. Thus, following our example, if the real difference in $\alpha$ and $\beta$ parameters between all choices in male-male pairs vs. male-female pairs was at the tails of the distribution of differences for shuffled pairs, the difference in parameters between groups was marked as significant. Importantly, this permutation test implementation assesses empirical significance (i.e. percentile of the reference permutation distribution) directly, and does not produce a test statistic or confidence interval.

Besides comparisons *between* groups, we were also interested to understand if certain groups stood out from the entire population. To assess the significance of $\alpha$ and $\beta$ parameter values per subgroup (that is, a one-sample test for a group), we compared these values to a reference bootstrap distribution for $\alpha$ and $\beta$, constructed from $N$ = 5000 randomly selected samples of $N$ = 279 participants taken (with replacement) from the entire population. The choices were again averaged within each sample and entered into the Fehr-Schmidt model. The resulting reference distributions were then consulted to check if $\alpha$ or $\beta$ values from specific subgroups fell outside the confidence intervals (95, 99, 99.9%) of these reference distributions, drawn from the entire study population, either above or below.

## Reporting summary

Further information on research design is available in the Nature Portfolio Reporting Summary linked to this article.

## Results

As a first step, we set out to replicate the well-established developmental trajectory where inequity aversion increases with age[13,14,16] and disadvantageous inequity aversion (DI) manifests stronger than advantageous inequity aversion (AI). In addition, we specifically added analyses that would test the effect of the gender of Allocator and Recipient on the rejection of unequal outcomes. In our experiment, each child made one choice (equal or unequal outcome) per choice dilemma. Thus, in order to analyse choices per dilemma as a function of age, we initially binned choices in 3 age groups (3–4yo; 5–6yo; 7–8yo; cf. 5 age-groups in Blake & McAuliffe[14]) to construct age-group averages of inequity rejection (and their bootstrapped standard errors). In the formal statistical analyses (see mixed-effects models below), Age in years was used as a continuous predictor instead and predicted effects of significant variables are visualised. As the focus of our experiments was to show potentially diverging patterns of choice according to the gender of both the Allocator and the Recipient, Fig. 2 shows the mean rejection rates per dilemma per age group and Dyad-gender combination.

What can be seen from Fig. 2 is that, descriptively, the proportion of inequity rejections (i.e. inequity aversion) increased with age across reward distributions (see Supplementary Tables S1–4 for the inequity rejection rates per subgroup). Regarding gender of the Allocator and Recipient, on a descriptive level, boys rejected unequal outcomes more often in both disadvantageous distributions when paired with a male (non-costly: 81.30%; costly: 62.50%) than when paired with a female Recipient (non-costly: 72.50%; costly: 59.40%). The reversed pattern was found in the two advantageous distributions, i.e. boys preferred choices with equal outcomes when paired with a female (non-costly: 58.00%; costly: 39.10%) compared to a male Recipient (non-costly: 48.40%; costly: 23.40%). Likewise, girls made often rejected inequity in both disadvantageous distributions when paired with a male (non-costly: 81.40%; costly: 64.30%) compared to a female Recipient (non-costly: 67.10%; costly: 48.70%). In all groups, there were always more inequity rejections in the non-costly compared to the costly distributions, suggesting that the outcome difference level or the cost of outcome inequity reduction influenced rejections. We therefore also modelled choice data with the Fehr-Schmidt model[5,13,33] (see below), where the subjective (dis)-utility of both own payoff and inequity is modelled simultaneously. On the whole, these results already preclude a simple efficiency maximising choice heuristic, as both boys and girls preferred the numerically inferior option (2 vs. 5 or 6 tokens) in the disadvantageous distributions. Similarly, an always-choose-equal rule cannot explain these results, either, because of considerable within-subject variability in choosing the equal option across choice options. Moreover, in at least the costly advantageous condition, both boys and girls (averaged across ages) did *not* prefer the equal outcome.

As each child was either first or second in the role of Allocator, we compared whether being Allocator first or second influenced choices over all dilemmas. We did not find statistically significant evidence that Allocator order impacted choice allocation in any of the dilemmas (Chi-square tests for equal proportions: DI-non-costly $\chi_{(1)} = 0.275$; $p = 0.600$; DI-costly $\chi_{(1)} = 2.398$; $p = 0.122$; AI-non-costly $\chi_{(1)} = 0.000$; $p = 1.000$; AI-costly $\chi_{(1)} = 0.005$; $p = 0.946$). As the Recipient was in the same room, but not facing the Allocator in our design, there was no immediate feedback from the Recipient on the Allocator. This is in contrast to the design introduced by Blake & McAuliffe[14,16], where the Recipient is facing the Allocator *during* the choices, and e.g. House et al.[17], who found strong effects related to laughter when the Recipient was present over when they were absent.

### Analysis of the choice data with mixed-effect models

Following Blake et al.[16] we used mixed-effect models ('lme4' package in R[69]) to estimate the effects of Age (continuous, in years), Type of Cost (Non-costly vs. Costly; referred to as CostType), Type of Inequity (DI vs. AI;

### Table 2 | Mixed-Effects model for Age, CostType and InequityType

| Baseline Model | Estimate | SE | Z-val | P value |
|---|---|---|---|---|
| Intercept | −3.39908 | 0.51422 | -6.61 | 3.84E-11 |
| Non-Costly | 0.90193 | 0.14089 | 6.402 | 1.54E-10 |
| Age (continuous) | 0.61587 | 0.08287 | 7.432 | 1.07E-13 |
| AI over DI | 1.06529 | 0.62116 | 1.715 | 0.08635 |
| Age × AI | −0.32607 | 0.09987 | −3.265 | 0.00109 |

Binary choices (1 per $N = 4$ conditions, $N = 279$ subjects per choice) are modelled within subjects, with Age as a between-subjects variable. Coefficients are shown in logits.

InequityType), Gender of Allocator (AllocatorGender) and Gender of Recipient (RecipientGender) on Equity Choice (that is, rejection of inequity and choosing the 1:1 alternative). We used forward model selection, based on Akaike Information Criterion and statistical comparison of nested models with Likelihood Ratio Tests, to arrive at the most parsimonious solution (see Table 2 for that model). Remember that rejecting an unequal distribution in our experiments did not result in zero payoff, but in the equal (1:1) alternative. That is, some rejections were non-costly, such as choosing the 1:1 alternative over the 1:4 disadvantageous distribution. For the non-costly inequity rejections, we thus expected to find a higher baseline of inequity rejection. Figure 3 shows the predicted effect of Age, Inequity Type and Cost Type on the proportion of inequity rejections (using the 'effects' package in R[70–72]).

The best model (using forward selection on the mentioned terms and their possible interactions) showed a significant effect of Age, CostType and an interaction between Age × InequityType (see Table 2 for coefficients and exact *p* values). The main effect of Age was replicated for AI and DI separately as well (Supplementary Tables S5–6) and for datasets split on CostType (Supplementary Tables S7–8, Supplementary Figs. S1–2). The interaction between Age × InequityType was found for the non-costly dataset, but not for the costly dataset. With the current dataset, we did not find statistically significant evidence that including an interaction between CostType × Age would improve the model fit (LRT, $\chi^2_1 = 0.6898$, $p = 0.406$), nor for including the interaction between CostType × InequityType (LRT, $\chi^2_1 = 0.1582$, $p = 0.691$), or a triple interaction between Age × CostType × InequityType (LRT, $\chi^2_2 = 0.9756$, $p = 0.614$). Furthermore, we found no statistically significant evidence that adding a factor indicating if Allocators had 1 or more siblings (74% yes, 26% no) would improve the model fit (LRT, $\chi^2_1 = 0.0517$, $p = 0.820$).

This baseline model thus replicates the significant effect of Age found in previous studies, and the interaction between Age and InequityType (DI vs. AI) as shown by Blake & McAuliffe[14] for age groups, and by Blake et al.[16] for continuous age (also when analysing just rejections of the unequal distributions in their published data; our re-analysis).

The bootstrapped choice data in Fig. 2 already shows interesting descriptive patterns when the choices are split according to gender of the Allocator and Recipient. Female Allocators seem to be more willing to reject inequity when facing AI, thus acting prosocial and (in the Costly dilemma) foregoing additional payoff to themselves, while Allocators paired with male Recipients seem to be more likely to reject DI outcomes that would favour these male Recipients. To test these observations, we added AllocatorGender, RecipientGender and possible interactions of these factors with InequityType to the baseline model (Table 2). Indeed, adding either AllocatorGender or RecipientGender as a factor interacting with InequityType significantly improved the model fit (LRT$_{Allocator}$, $\chi^2_2 = 9.6085$, $p = 0.0082$; LRT$_{Recipient}$, $\chi^2_2 = 16.737$, $p = 0.0002$, Tables 3–4).

Additional models exploring the addition of interactions with CostType were fitted, but we did not find statistically significant evidence for increased model fit by including such interactions (see Supplementary Tables S9–10). Figures 4 and 5 show the predicted effects for the InequityType × Gender interaction (Fig. 4: AllocatorGender; Fig. 5:

**Fig. 3 | Predicted effect of Cost Type on inequity rejection.** Figure shows the predicted effects of the Age × InequityType interaction, separated by Cost type. Shading in predicted effect plots always indicates 95% CIs. AI advantageous IA, DI disadvantageous IA. Inner ticks indicate individual data points. $N = 279$, as all predicted effects displayed are within-subjects.

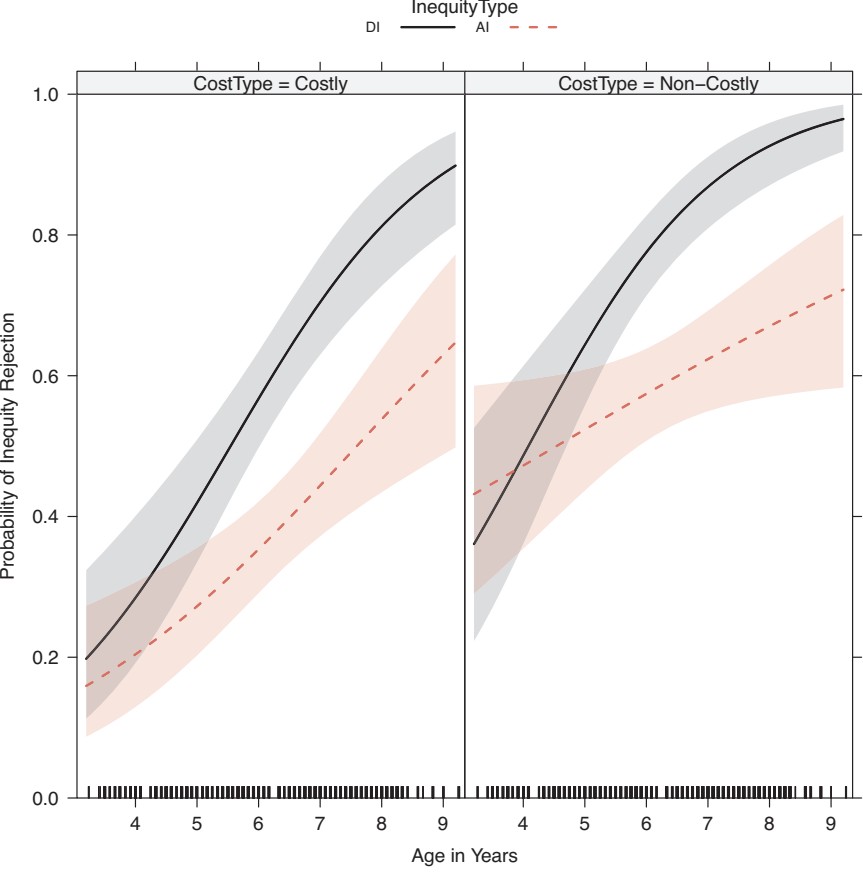

## Table 3 | Mixed-Effects model including Allocator Gender

| Model including Allocator Gender | Estimate | SE | Z-val | P value |
|---|---|---|---|---|
| (Intercept) | −3.56 | 0.53 | −6.67 | 0.00000 |
| Age (continuous) | 0.62 | 0.08 | 7.45 | 0.00000 |
| Non-Costly over Costly | 0.91 | 0.14 | 6.43 | 0.00000 |
| AI over DI | 1.47 | 0.64 | 2.29 | 0.02180 |
| Age × AI | −0.33 | 0.10 | −3.30 | 0.00097 |
| Male Allocators | 0.26 | 0.22 | 1.17 | 0.24377 |
| AI × Male Allocators | −0.80 | 0.28 | −2.87 | 0.00407 |

Binary choices (1 per $N = 4$ conditions, $N = 279$ subjects per choice) are modelled within subjects, with Age as a between-subjects variable. Coefficients are shown in logits.

## Table 4 | Mixed-Effects model including Recipient Gender

| Model including Recipient Gender | Estimate | SE | Z-val | P value |
|---|---|---|---|---|
| (Intercept) | −3.91 | 0.55 | −7.11 | 0.00000 |
| Age (continuous) | 0.64 | 0.08 | 7.61 | 0.00000 |
| Non-Costly over Costly | 0.92 | 0.14 | 6.45 | 0.00000 |
| AI over DI | 1.80 | 0.66 | 2.74 | 0.00621 |
| Age × AI | −0.36 | 0.10 | −3.54 | 0.00040 |
| Male Recipients | 0.73 | 0.23 | 3.19 | 0.00141 |
| AI × Male Recipients | −1.13 | 0.28 | −4.00 | 0.00006 |

Binary choices (1 per $N = 4$ conditions, $N = 279$ subjects per choice) are modelled within subjects, with Age as a between-subjects variable. Coefficients are shown in logits.

RecipientGender), depicted at various levels of Age and visualised split by CostType. With the current dataset, we did not find statistically significant evidence for an increase in model fit when including a triple interaction with continuous Age (InequityType × AllocatorGender × Age; InequityType × RecipientGender × Age).

### Direct comparison with the dataset from Blake et al. (2015, Nature)

Thus, in contrast to reports from studies by Blake, House and co-authors[16,31], we found evidence that the gender of both Allocators (Table 3) and Recipients (Table 4) plays a role in decisions to accept or reject unequal outcomes. To dig deeper into why our results might differ from previous reports, we included the raw data from ref. 16 in our analyses for direct comparison. Given that our study population comes from a WEIRD (Western, educated, industrialised, rich and democratic) population (urban, middle- to upper-middle class areas in Düsseldorf, Germany), we compared our results to their Canadian and US samples. For comparison, it is important to note that the costly AI distribution in our experiment was 2:0, compared to 4:1 in Blake et al., and Blake only included FF and MM gender dyads. The raw choices for the costly DI and AI dilemmas are depicted in Supplementary Fig. S3, where a similar trend of diverging choices by Gender can be seen. As the raw inequity rejection rates are likely influenced by experimental design considerations, we can also directly compare to predicted rejection rates from the GLMMs. Figure 6 shows these, as calculated for the FF and MM dyads in both datasets (limited to the age range 3–9). What immediately stands out is the strong interaction between GenderAllocator and Age that is present in our AI data (see Table 3), but not in the Blake dataset (see however Supplementary Table S11 and Supplementary Fig S4 for a significant Age × GenderAllocator effect in the Blake dataset across the full Age range).

**Fig. 4 | Predicted effect of Allocator Gender on inequity rejection.** Figure shows the predicted effects of the InequityType × GenderAllocator interaction, over brackets of Age in years. Top row: predicted effects of this interaction for costly rejections. Bottom row: predicted effects of this interaction for non-costly rejections. Error bars indicate 95% CI. Columns: age bins of 1 year (lower bound given). Sample size Age bins: 3yo $N = 17$; 4yo $N = 48$; 5yo $N = 54$; 6yo $N = 58$; 7yo $N = 64$; 8yo $N = 38$. Total $N = 279$.

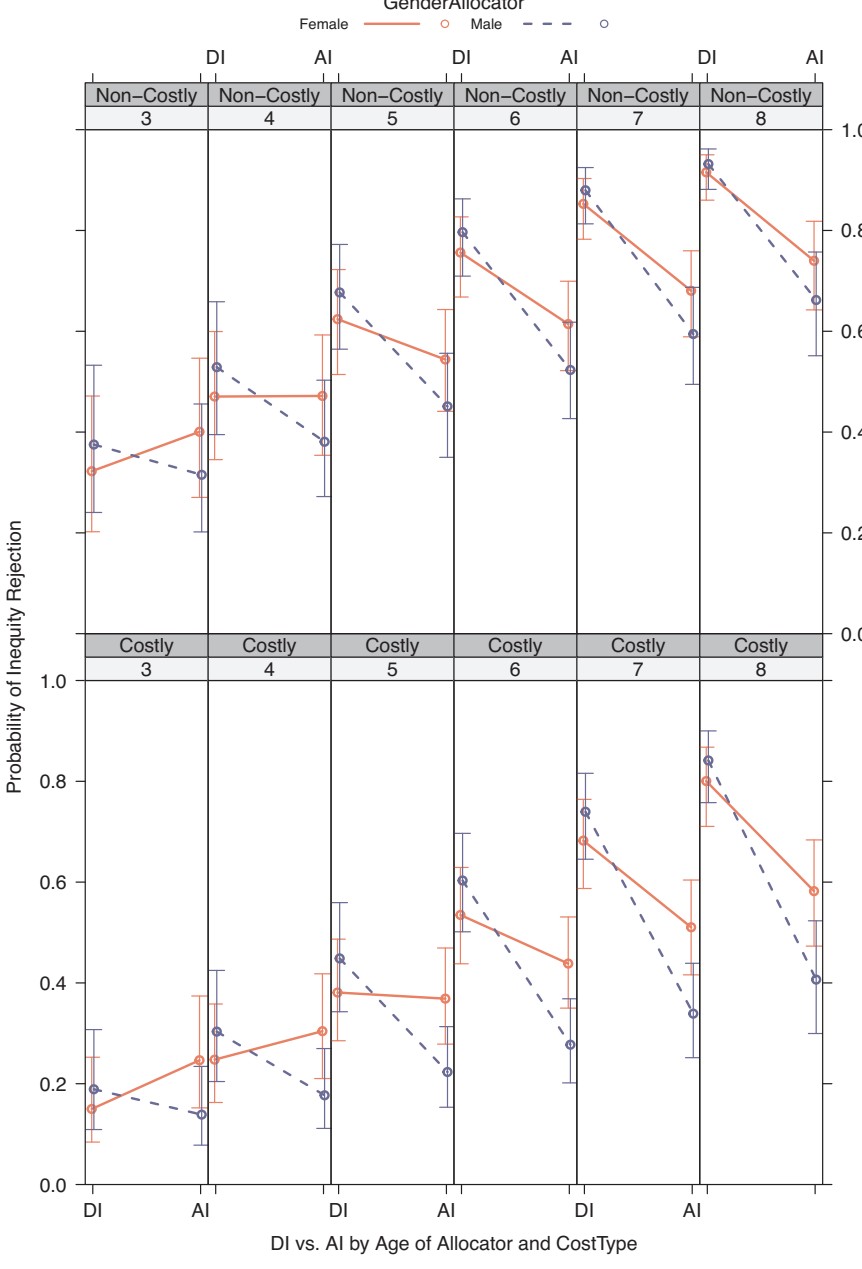

### Model-based analysis of gender-dyad dependent inequity aversion

As we have shown in our results, the explicit offer of an equal-outcome alternative can induce differences in inequity aversion, further determined by the costliness of such a choice. In order to get a better understanding of the differences in subjective utility that might underlie these social preferences, we fitted the Fehr-Schmidt model of inequity aversion[5] to the raw choice data. The Fehr-Schmidt-model reduces the model complexity and yields quantitative parameter estimates for $\alpha$ (DI) and $\beta$ (AI, see 'Methods').

Briefly, to estimate variability in $\alpha$ and $\beta$ for a given subgroup, we applied a bootstrap approach with resampling, essentially repeating the modelling step for a randomised subsection of the original group and aggregating the obtained $\alpha$ and $\beta$ values in a distribution from which we report the mean and variance (see 'Methods' for details). Subgroup scores were assessed for significance in comparison to confidence intervals on a reference population acquired similarly through bootstrap resampling of the original complete dataset. The 95th, 99th and 99.9th percentile confidence

intervals on these distributions are represented by dashed grey lines in the figure panels below (Figs. 7–8). Comparisons between subgroups were assessed for significance through bootstrap permutation analyses (see Supplementary Fig. S5 for a sensitivity analysis).

### Inequity aversion increases with age

As a first step in our model-based analysis, we confirmed our report above that inequity aversion (both DI and AI) increases with age in young children. The significant increase in $\alpha$ (e.g. DI) is already visible between ages 3–4 vs. 5–6 (uncorrected $p = 0.0004$; corrected $p = 0.0024$, $N = 5000$ bootstrap permutation test with Bonferroni correction for multiple comparisons, see Fig. 7), whereas a significant increase (uncorrected $p = 0.0016$; corrected $p = 0.0096$, permutation test) in $\beta$ shows up only in the transition from 5–6yo to 7–8yo children (see Supplementary Table S12 for parameter estimates per subgroup). We interpret this as evidence that DI develops earlier than AI, in line with the results from the GLMMs (significant effect of Age, significant interaction between Age × InequityType) and replicating

**Fig. 5 | Predicted effect of Recipient Gender on inequity rejection.** Figure shows the predicted effects of the InequityType × GenderRecipient interaction, over brackets of Age in years. Top row: predicted effects of this interaction for costly rejections. Bottom row: predicted effects of this interaction for non-costly rejections. Error bars indicate 95% CI. Columns: age bins of 1 year (lower bound given). Sample size Age bins: 3yo $N = 17$; 4yo $N = 48$; 5yo $N = 54$; 6yo $N = 58$; 7yo $N = 64$; 8yo $N = 38$. Total $N = 279$.

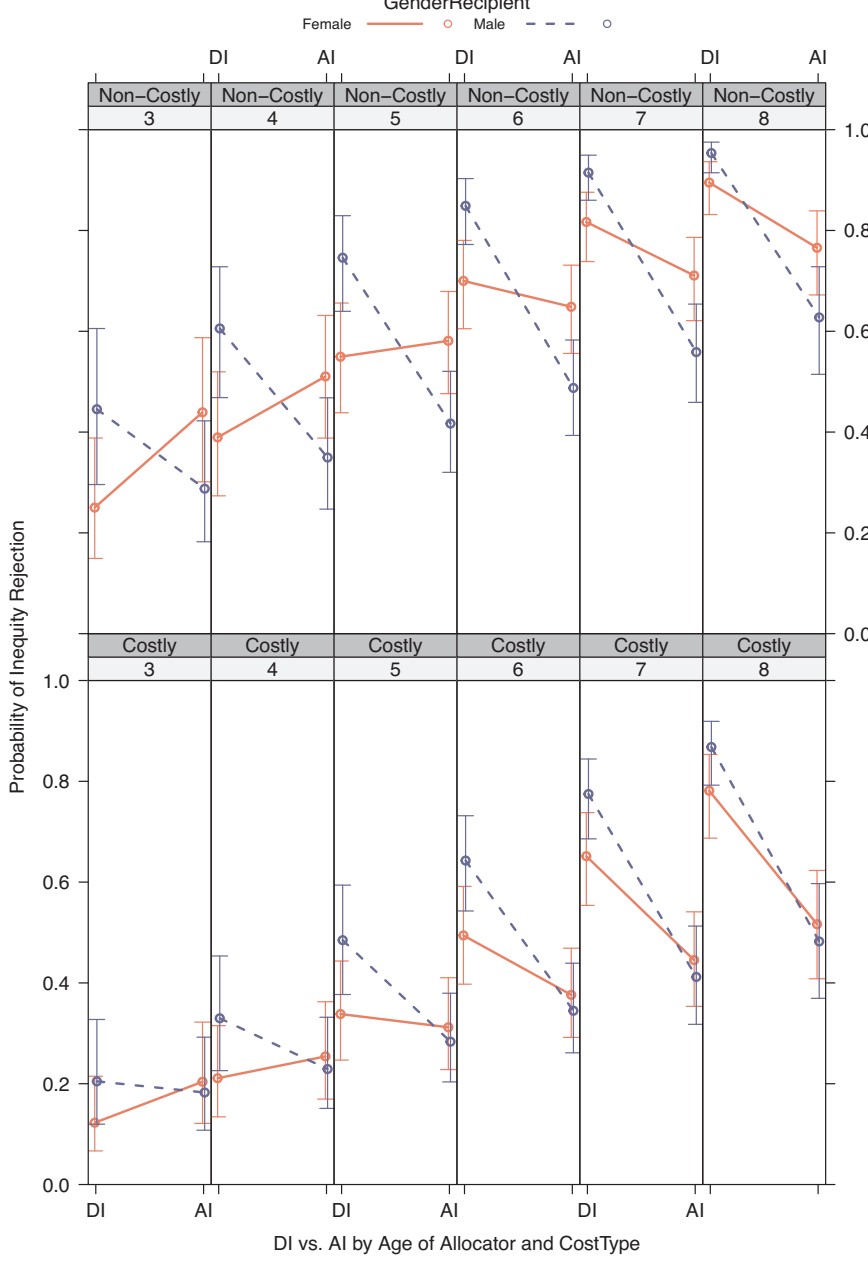

previous reports[13,14,16,33]. We found that the estimate for the $\beta$ parameter in the 5–6yo age group was significantly lower than the reference population (i.e. all children combined) average ($p < 0.001$ in a one-sample Z-test: outside the 99.9% of the reference bootstrap distribution). This effect was already found in other studies[34,35] and might be due to the fact that the development of AI ($\beta$) includes overcoming an initial spiteful preference for diminishing others' relative payoff. While $\beta$−levels did not significantly differ between 3–4yo and 5–6yo children, they increased significantly from 5–6yo to 7–8yo children (see above) to levels significantly higher than average ($p < 0.001$ in a one-sample Z-test).

## Gender and Dyad-gender-dependent differences in inequity aversion

To investigate gender-differences in IA, we re-organised our sample according to the gender of the Allocator (child making the decision) and the Recipient (partner). We found that parameter estimates for $\alpha$ did not differ

between subgroups of female or male Allocators (Fig. 8A) and estimates for $\alpha$ parameters for both groups fell within the reference distribution. However, parameter estimates for $\beta$-values for female Allocators were higher than the reference distribution (higher AI; $Z = 2.3433$; $p = 0.026$, one-sample Z-test), and parameter estimates for $\beta$ values for male Allocators were lower (lower AI; $Z = -2.876$; $p = 0.006$) than average. Moreover, a pairwise comparison revealed significantly higher estimates for the $\beta$ parameter in subgroups of female compared to subgroups of male Allocators (uncorrected $p = 0.0156$; corrected $p = 0.0312$, Fig. 8B). When the *Recipients* were male, the $\alpha$ estimate for their subgroup was higher than average ($Z = 2.323$; $p = 0.027$), although when Recipients were female, the $\alpha$ estimate for their subgroup fell on the border of the reference distribution. To check if DI was rejected more when Recipients were male than when they were female, a permutation test indeed revealed significantly higher $\alpha$ estimates for subgroups with male Recipients compared to subgroups with female Recipients (uncorrected $p = 0.016$; corrected $p = 0.032$, Fig. 8A).

**Fig. 6 | Comparison of predicted effects on inequity rejection with data from ref. 16.** Figure shows the predicted effects of the interaction between Age and GenderAllocator on Inequity Rejection. Fits are made separately on our data (FF vs MM only) and the Blake dataset. Model fits are shown, for costly choices only (our data) and WEIRD children ages 4–9 only (Blake data). Dashed horizontal lines indicate predicted inequity indifference (50% inequity rejections). Datapoints are $N = 50$ interpolated points on the predicted effect curve. 95% CI shading omitted for clarity. FF Female Allocator with Female Recipient $N = 76$, MM Male Allocator with Male Recipient $N = 64$.

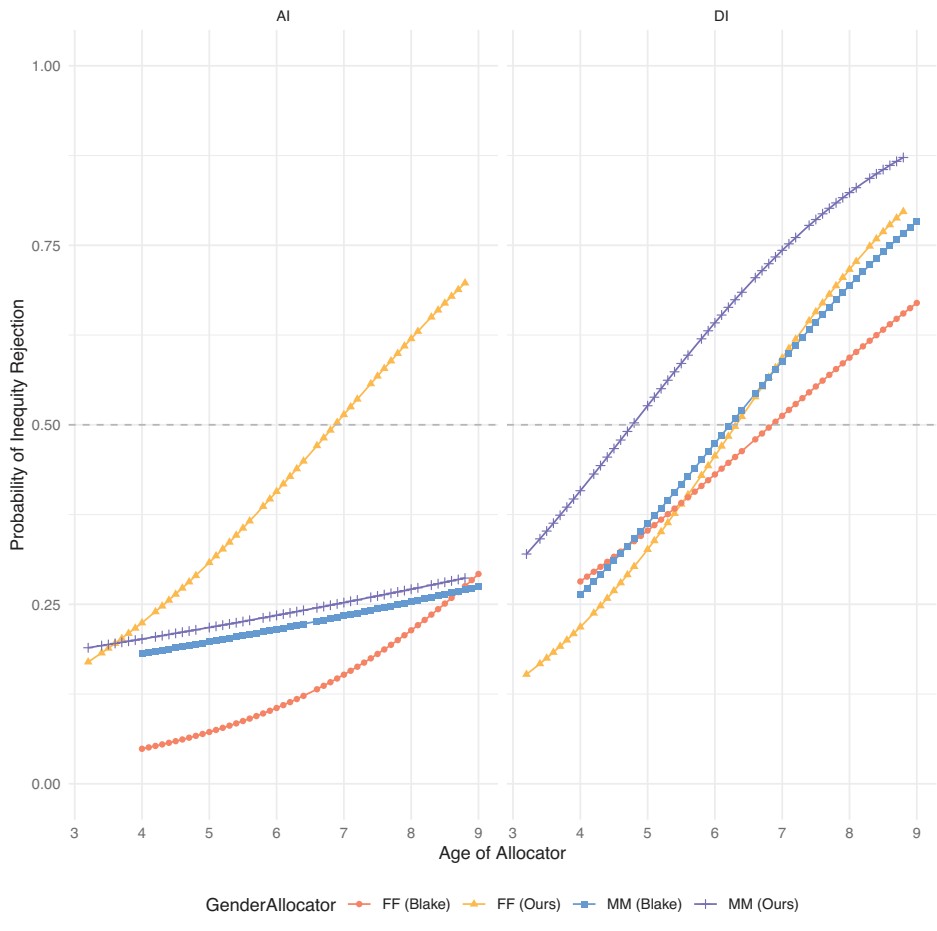

Estimates for the $\beta$ parameter broken down by subgroups of Recipient fell within the reference distribution.

Unpacking these main effects of gender, we found that Allocators did not treat all Recipients equally. Indeed, the observed lower $\alpha$ parameter for subgroups of female Recipients originated mostly from Allocators who were also female: while the estimates for the $\alpha$ parameter in male Allocators showed similar and average values for subgroups paired with Recipients of either gender, the estimates for the $\alpha$ parameter in female Allocators showed markedly lower values (compared to the reference distributions) in subgroups with female Recipients ($Z = -3.026$; $p = 0.004$, one-sample Z-test), and higher values in subgroups paired with male Recipients ($Z = 2.733$; $p = 0.009$) compared to the reference distribution. This suggests that female Allocators showed markedly less DI (i.e. allowed their partners to be better off) with female Recipients compared to male Recipients. suggesting that female Allocators tolerated being worse off than the Recipients better when the Recipients were female than when they were male. Thus, girls showed an envy bias as they were selectively more generous with other girls than with boys ($p < 0.001$ permutation test, Fig. 8C). Conversely, the $\beta$ parameter estimates for girls did not differ significantly for subgroups of female Allocators paired with either male or female Recipients. The significant difference in estimates for $\beta$ parameters between male and female Allocators was primarily driven by the difference in $\beta$ parameters for male Allocators, showing a descriptive spite gap: higher ($Z = 2.02$; $p = 0.043$) estimates for subgroups paired with female Recipients vs. much lower ($Z = -5.086$; $p < 0.001$) estimates for subgroups paired also with male Recipients. Our design was not powered statistically enough to support this trend statistically with a pairwise comparison (but see Fig. 9 for a partner contrast analysis by Age group).

### Development of Dyad-gender-dependent differences in fairness preferences

Finally, we asked how these diverging patterns in egalitarianism relative to Allocator- and Recipient gender manifested across the age groups. We focused on the strongest effects from the analyses that so far excluded age: the envy-bias, i.e. the girls' tendency to be more tolerant toward Recipients being better off when Recipients were female compared to male, and the spite gap, i.e. subgroups of boys showing lower (sometimes even strongly negative; see Supplementary Table 12 for 5–6yo MM pairs) estimates of $\beta$ parameters when paired with other boys, compared to subgroups of (generous) female Allocators paired male Recipients. Indeed, we found that the envy bias in female Allocators against cross-gender Recipients showed a trend of increasing values with Age group. While we observed no significant envy bias in female Allocators in the youngest Age group, it appeared in the middle Age group ($p = 0.0336$, one-sample Z-test on the Recipient Contrast for female Allocators bootstrapped distribution against 0) and peaked in the Age group with 7–8yo children ($p = 0.0044$, Fig. 9A). In a similar vein, the gender spite gap of boys against other boys, in comparison with female Recipients, statistically manifested significantly only in the oldest age group ($p = 0.0364$, one-sample Z-test on the Allocator Contrast for male Recipients bootstrapped distribution against 0; Fig. 9B). Note here that our analyses do not speak to a change in either the envy bias or spite gap across age groups, but merely to the statistically detectable presence of the effect per age group.

### Discussion

With this study, we provide a replication of the findings that both advantageous inequity aversion (AI) and disadvantageous inequity aversion (DI) increase with Age, and that children (from WEIRD backgrounds) reject DI

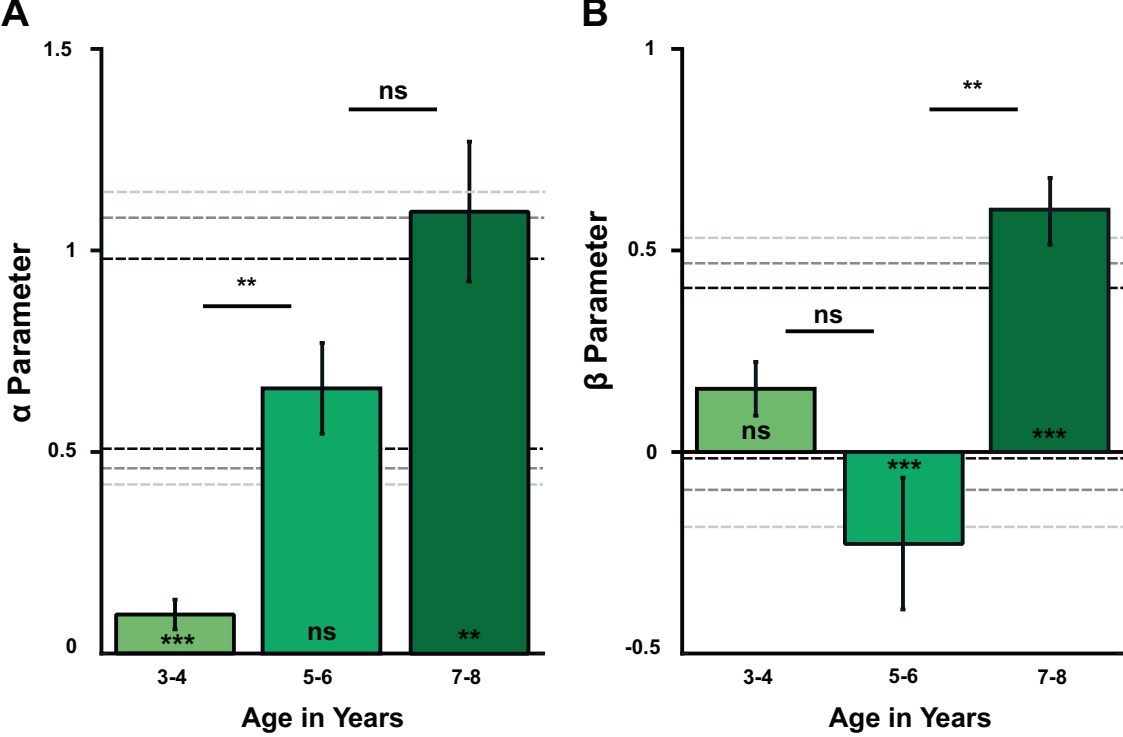

**Fig. 7 | Development of α- and β-parameters with age.** Development of α- (**A**) and β-parameters (**B**) with age (bootstrap means ± bootstrapped standard error). Between-group differences were assessed for significance using a pairwise permutation approach, with significance levels determined from the empirical permutation distributions and adjusted with Bonferroni-correction for multiple comparisons. One-sample Z-tests for significance (stars above x-axis) reflect comparisons to the confidence intervals of the global resampled parameter distributions. *: $p < 0.05$; **: $p < 0.01$; ***: $p < 0.001$. Sample sizes: 3–4yo $N = 65$; 5–6yo $N = 112$; 7–8yo $N = 102$

stronger than AI. In addition, we provide evidence through a linear mixed-modelling approach that egalitarian preferences were also significantly influenced by the gender of both the Allocator and the Recipient in the interacting dyad, and that a similar gender effect can also be detected in a comparable subgroup of a previous study. Using a subjective utility modelling approach, we found that girls were more generous by rejecting AI more than boys, and Allocators paired with boys rejected DI more often than Allocators paired with girls. Notably, this reduced rejection for DI originated from girls, who exhibited an *envy bias*, i.e. they were generous, rejecting DI less when other girls would be better off, compared to when they were partnered with boys. We also observed a *gender spite gap* in the oldest Age group, revealing that these boys treated other boys with spite (advantageous inequity non-aversion, i.e. they attached positive value to being better off than other boys), and did so significantly more than a subgroup of male Recipients paired with female Allocators. Finally, girls revealed unconditional AI, i.e. there was no difference in AI rejection rates when the Recipient was female or male.

### Related literature
In psychology, behavioural economics and related fields, there is a growing interest in understanding the development of social preferences in childhood. The general finding across nearly all studies is that older children are less selfish and more prosocial than younger ones. For example, older children share more resources in dictator (or similar) games[41,45–47,57], become increasingly concerned with social norm compliance and norm enforcement[42,51,59] and show more direct and indirect reciprocity[65]. Accordingly, children reveal progressively more egalitarian preferences between the ages 3–8[13–16,32–35], although there seem to be cultural differences in the emergence of advantageous inequity aversion[16,17,31]. In addition, evidence also suggests that preferences for equal resource allocations are gradually replaced by other, more complex social preferences in children and adolescents beyond this age range, such as efficiency seeking[30,33,41],

meritocratic fairness views[41], social welfare considerations[30,39], reputation concerns[58], maximin preferences, i.e. the desire to increase the minimum payoff in a group[30] and parochial altruism and intergroup biases[33]. Furthermore, even though disadvantageous inequity aversion in younger children might be driven by spite[35], the spiteful tendencies reported here and in other studies[33,35] seem to decrease between age five and ten[34].

Several studies controlled for and/or reported the role of gender in the development of social preferences. Gummerum et al.[46]. found that young girls aged three to five were more generous in the dictator game than boys. Consistent with this, Deckers et al.[44]. observed that girls of age eight or nine were more altruistic than boys and were significantly more likely than boys to choose an equal split allocation in an allocation game. Harbaugh, Krause and Liday[47] discovered that, in children aged 7 to 18, girls were less selfish than boys and made larger offers in the dictator and ultimatum games, although Harbaugh, Krause and Vesterlund[48] did not find gender differences in the ultimatum game in children of the same age range. Fehr, Glätzle-Rützler and Sutter[33] reported that girls aged 9 to 17 were more egalitarian than boys, but also less altruistic, but there were no gender differences in spite. LoBue et al.[32] investigated emotional and behavioural signs of inequity aversion in children aged three to five and observed a weak and statistically non-significant trend in boys to respond to inequity with more unhappiness and more dissatisfaction than girls. Almas et al.[41] investigated children in school grade levels 5 to 13 and reported no gender effects in young children, gender effects only started to emerge in adolescence; male adolescents were more efficiency-oriented than female adolescents. McAuliffe, Jordan and Warneken[51] investigated third-party punishment in 5–6-year-old children and report that boys were more likely than girls to punish inequity. Fehr et al.[13]. found that boys aged three to eight showed stronger parochial tendencies than girls and were less averse against disadvantageous inequity if the partner was an ingroup member.

Far fewer studies set themselves the explicit goal of investigating the role of gender in the development of social preferences. Of the few notable

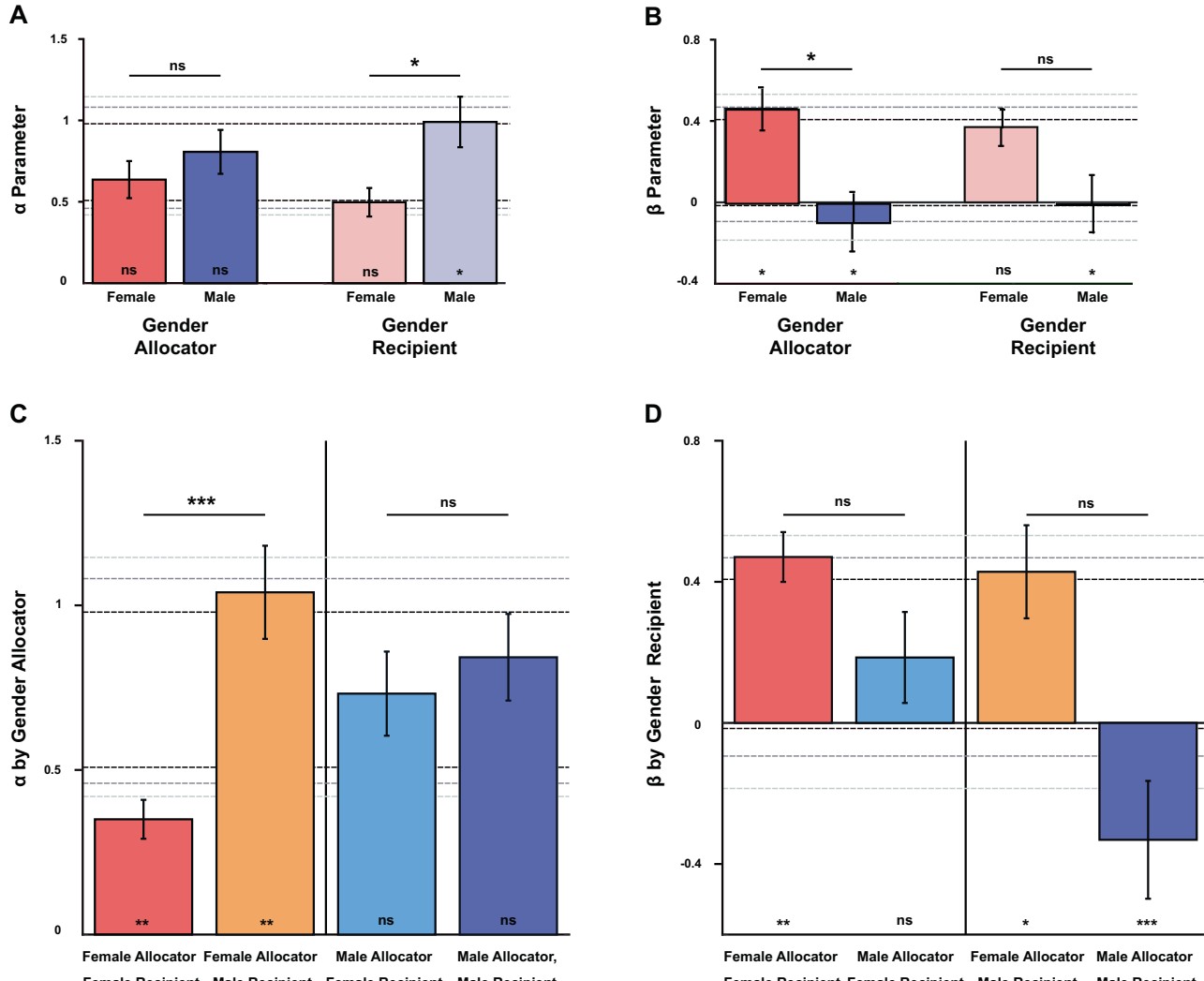

**Fig. 8 | Effects for Allocator-, Recipient- and Dyad-gender on $\alpha-$ and $\beta-$parameters.** Figure shows effects for Allocator and Recipient gender and for $\alpha$ (disadvantageous IA; **A**) and $\beta$ (advantageous IA; **B**), and differences in $\alpha$ (**C**) and $\beta$ (**D**) split by gender of Allocator and Recipient. Bars indicate the mean of the bootstrapped population scores ($N = 5000$ draws). Error bars indicate the standard deviation of the bootstrapped population and thus act as s.e.m. Significance between groups was assessed through bootstrapped permutation tests with Bonferroni correction for multiple comparisons. One-sample $Z$-tests for significance (stars above x-axis) reflect comparisons to the confidence intervals of the global resampled parameter distributions. \*: $p < 0.05$; \*\*: $p < 0.01$; \*\*\*: $p < 0.001$. Sample sizes: Female Allocators $N = 146$; Male Allocators $N = 133$; Female Recipients $N = 145$; Male Recipients $N = 134$; Female Allocators $\times$ Female Recipients $N = 76$; Female Allocators $\times$ Male Recipients $N = 70$; Male Allocators $\times$ Female Recipients $N = 69$; Male Allocators $\times$ Male Recipients $N = 64$.

exceptions, Martinsson et al.[39]. revealed in children between 10 and 15 years that competitive preferences and self-interest was not different between genders, but girls were more inequity averse than boys, especially in disadvantageous situations, and cared less for social welfare, although social welfare considerations gained in importance with age in all genders. Benenson and colleagues report more egalitarian choices in a resource sharing task in 3–5-year-old girls than boys[73]. Sutter et al.[30]. studied differences in the development of social preferences in boys and girls aged 8 to 17 years. They found that, in boys, efficiency concerns became more important with age, while inequity aversion lost importance. By contrast, in girls, maximin preferences, i.e. the desire to increase the minimum payoff in a group, became more important with age and in children older than 12 years, girls were more inequity averse than boys. Finally, it is important to note that several studies found no, or no consistent, gender effects in social preferences during development[14,16,31,45,48,57,59,63].

Thus, in summary, there is considerable heterogeneity in the direction of the reported gender effects, or whether gender effects were found at all. The general tendency in those studies that do find gender effects seems to be

that girls are more prosocial and boys more competitive. All studies reviewed in this section that report gender effects on social preferences considered the gender of the allocating child; it remained an open question whether the gender of the receiving child and/or the gender composition between allocating and receiving children (in dyadic or triadic interactions) influences fairness preferences.

### Fairness preferences are not a function of gender per se, but of the Dyad-gender composition
Our data may reconcile some of the heterogeneity in the literature in the direction of gender effects in the ontogeny of social preferences, or whether gender effects were found at all. As outlined in the previous paragraph, gender effects, if they were investigated or reported at all, only concerned the gender of the Allocator; the gender of the Recipient was not considered or examined, and it was often not even controlled for, e.g. in anonymous interactions. In our study, we show that fairness preferences are not a fixed function of the Allocator's gender, but critically depend on the Dyad-gender composition. Thus, depending on whether they are paired with a male or

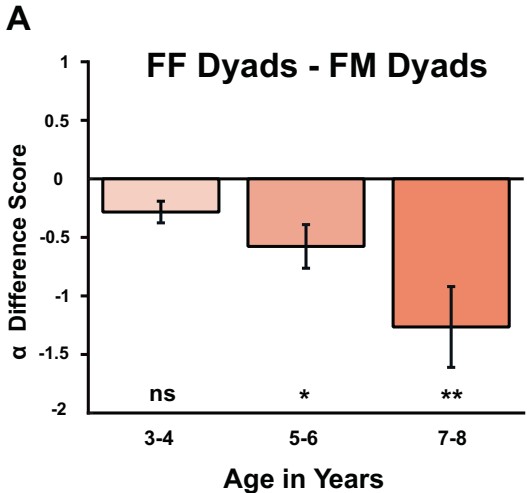

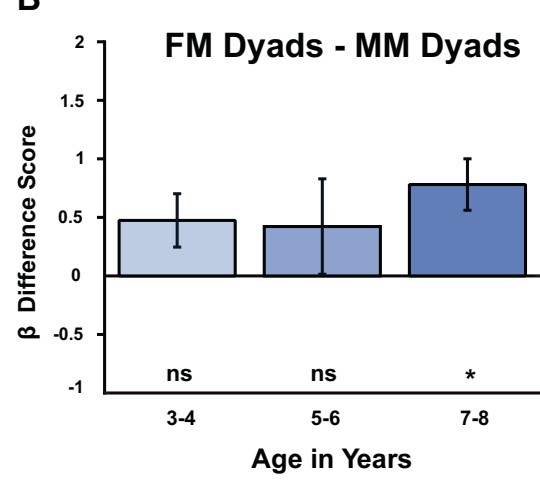

**Fig. 9 | Development of envy bias and spite gap.** Panels show the difference in parameter estimates for female Allocators towards female minus male Recipients (**A**) or for female minus male Allocators towards male Recipients (**B**). Female Allocators exhibited lower levels of DI ($\alpha$-parameter) against female Recipients compared to male Recipients (**A**). The difference in AI ($\beta$-parameter) between female vs. male Allocators towards male Recipients, is significant only in the group with the oldest children (**B**). Scores indicate the difference between two bootstrap populations with respect to Recipient gender (**A**) or Allocator gender (**B**). Error bars indicate the standard deviation of the difference between these contrasted populations across bootstraps. Significance levels were assessed with a permutation analysis at the level of Allocator (**A**) or Recipient (**B**) per Age group as indicated. *: $p < 0.05$; **: $p < 0.01$. Sample sizes: panel a (Female Allocators)—3–4yo $N = 32$; 5–6yo $N = 56$; 7–8yo $N = 58$; (b) (Male Recipients)—3–4yo $N = 36$; 5–6yo $N = 50$; 7–8yo $N = 48$.

female Recipients, boys or girls in the role of the Allocator may be more or less inequity averse, or, by extension, perhaps also more or less prosocial.

At least two of our Dyad-gender conditions have been reported before (girl-girl and boy-boy interactions[16,58,59]). We re-analysed the results published by Blake and colleagues using their publicly available dataset[16,74], and compared their findings to ours after limiting our dataset to girl-girl and boy-boy dyad conditions. We were able to replicate the results reported by Blake and colleagues (for their US and Canadian samples combined) with regard to the main effect of Age (limiting both samples to a maximum Age of 9 years for comparability) on inequity rejection rates, and the difference between DI and AI conditions. This implies that the children's behavioural pattern in our dataset was likely not different from the pattern found in the dataset used by Blake and colleagues[16]. However, there was one notable difference between our data and those of Blake et al. while we found a strong increase in advantageous inequity rejections with age in female-female dyads, this gender effect in AI was not reported in the dataset of Blake et al. However, when we re-analysed their data across the full Age range, a similar GenderAllocator effect was observed (also, an additional interaction with Age appeared; see Supplementary Fig. S4 and Supplementary Table S11). Perhaps such an interaction with Age would also have shown for German children, had we included ages up to 12yo as in Blake. Of note is that, while the design of Blake et al., was also meant to uncover differences due to cultural/economical background, their sample size per country is lower than in our design and the significant gender effect in their data only shows up after combining the Canadian and US children. Although methodological differences remain, the generally high similarity in results across datasets is encouraging: it supports not only the confidence in the general validity of our and other results, but also the possibility mentioned above that a similar effect of Dyad-gender composition could have also been found in previous data if these conditions had been taken into account. We therefore argue that generalised statements about the role of gender in the development of social preferences, or the lack of such a role, should be taken with caution if the gender of all interacting parties was not considered or controlled for, as we and others[17] have shown that even highly situational factors can influence decisions about unequal outcomes. An interesting avenue for future research would be to run a meta-analysis on fairness preferences with the available datasets in which the genders of both players are considered.

## Interaction experiences and gender stereotypes

Our data offer further insights into the development of fairness preferences and, in particular, into the emergence of gendered biases in these preferences. One obvious explanation for our age effects on equity preferences is maturation of social cognition. That is, cognitive processes that play a role in social decisions, such as theory-of-mind and perspective-taking, might be better developed in older compared with younger children[75], enabling the older children to better anticipate the Recipients' expectations. Interestingly, the development of social cognition coincides with the emergence of children's ability to recognise and consider group delineations, group concerns and group-specific social norms[76,77]; it, hence, concurs with the rise of intergroup biases[33]. It is, therefore, possible that children not only start to develop their own gender identity with age, but also identify as being part of their own gender group as well as not being part of the opposite-gender group. This development of gender-group-membership identifications may account for the age-dependent differences in same-gender versus opposite-gender fairness attitudes. General maturation of social cognitive capacities might explain the age-related changes and perhaps even the differences in fairness preferences between girls and boys (assuming a diverging path in this maturation[78]). But how do children acquire differing same-gender versus opposite- gender fairness attitudes? Why do boys act predominantly competitive when interacting with other boys, but not with other girls and why are girls more compassionate when interacting with other girls? Reinforcement learning could provide clues.

Reinforcement learning[79] and evolutionary models of cooperation[80,81] predict that social intuitions are partly fashioned by past interaction experiences[1,40,81]. According to this idea, children gradually acquire a social response pattern that reflects their previous positive and negative experiences with other children in social exchange situations[40,48]. Since social attitudes differ between genders, Allocators' fairness preferences should be congruent with the current interaction Recipient's expected fairness attitude, dependent on the Recipient's gender and as learned by experience (e.g. female compassion, male competitiveness[30,82]). For instance, experienced or predicted male DI should be met with similar DI (e.g. envy is met with envy) and experienced or predicted female AI should be met with equal AI (e.g. compassion is met with compassion). This explanation is supported by our observation that the children's fairness attitudes were generally aligned to

that of their interaction Recipient, dependent on his or her gender, as demonstrated by the girls' envy bias, or the boys' persistent preference for advantageous inequity, when rejecting it would be costly, paired with other boys but not when paired with girls (see again Fig. 2, costly AI).

Importantly, however, girls also showed unconditional AI; that is, their intolerance toward being better off than others was not different between interactions with female and male Recipients. The fact that girls did not adjust their AI downwards when interacting with boys (who, on average, exhibited significantly lower levels of AI) is difficult to reconcile with a pure experience-based model of social preferences. Instead, this mismatch between social preferences and interaction experiences seems more consistent with the notion that fairness preferences were also shaped by elements beyond experience, as for instance by a compliance to social gender norms that could dictate unconditional compassion even with non-reciprocating partners. This would imply that gendered stereotypes of social behaviour could, under some circumstances, work against settling mutual social preferences on levels matching interaction experiences[36–38,83,84]. Especially in cross-gender interactions, social norms prescribing, for example, female unconditional kind-heartedness, or being 'nice and sweet'[83], could inflate female compassion to levels divorced from those expressed by males in these interactions[27,85]. A putative influence of such internalised social roles is also supported by the fact that effects of Dyad-gender-composition were weak or absent in the very young children of our sample but only showed up later in childhood. Though we did not measure gender stereotype endorsement in our participants, the emergence of such stereotypes could influence the concurrent development of fairness preferences. Thus, summarising, our data suggest that the emergence of gendered fairness preferences in childhood is unlikely the consequence of a single developmental process alone. Instead, the pattern of our children's egalitarian choices seems to reflect a mix of cognitive maturation, past interaction experiences and acquired social gender norms.

### Biological pre-disposition is an unlikely mechanism for Dyad-gender-dependent social preferences

Another possibility is that the development of these Dyad-gender-specific fairness preferences are not due to accumulating experience or the influence of gender role stereotypes at all, but the result of a biological pre-disposition that becomes gradually expressed with age. In consonance with this idea, differences between the sexes in adult decision making have often been explained with reference to natural selection in evolution[86]. Indeed, in our study, the observed higher compassion in females as well as higher competitiveness among males could reflect differences in the challenges faced by different sexes in the course of evolution. Females may have benefitted from the display of greater altruism towards non-kin to facilitate cooperative breeding and allomaternal care[87] and shield against potential conflicts, whereas males had to compete with other males for access to limited resources, including mating opportunities with females, and act as protectors against male enemies[81]. Thus, it might well be that such evolutionary pressures promoted higher levels of compassion in females and higher levels of competitiveness—leading to envy or even spite—in males. Precursors of these tendencies already could have manifested in our sample of young children and further develop with age.

However, though it is safe to say that biological sex plays a role in social preferences and decision making, it cannot fully explain the findings reported here. We found a double dissociation of partner-gender sensitivity in the current experiment: girls only differentiated between male and female partners within the 'typical male' competitiveness/envy context (disadvantageous distributions), showing significantly more generosity when paired with gender-matched partners. By contrast, boys were treated differently within the 'typical female' compassion context (advantageous distributions): female actors showed high compassion to boys, while male actors showed dramatically lower levels of compassion and even spite when paired with partners of the same gender. In addition, there is no obvious reason why such behavioural patterns should change during development in a phase where sex hormones and putative partner selection do not yet play

a major role. Instead, we consider it more plausible that the social choice patterns reported here are likely the result of acquired gendered social roles that reinforced, or came to override, past interactive experiences[21].

### Demographics cannot sufficiently explain gender-related differences in inequity aversion in this sample

Demographic factors, such as socio-economic status, number and gender of siblings, or birth order have been shown to influence egalitarian preferences[13]. However, our sample was recruited in middle- to upper-middle class urban childcare facilities and was, thus, socio-economically homogeneous. In addition, while socio-economic status, siblings or birth order might relate to fairness attitude as a (relatively stable) personality trait (but see ref. 50), these factors are not easily suited to explain our children's flexible, dyad-dependent adjustment of egalitarian preferences to the gender of the interaction partner. We, therefore, consider it unlikely that socio-economic status, birth order or siblings explain the Dyad-gender-dependency of egalitarian preferences found in our children (note that we did not record the birth order of our participants, so we cannot rule out this possibility with certainty; the factor of having 1 or more siblings did not significantly improve the baseline model fit). Finally, one study showed an effect of whether there was laughter in the face-to-face interaction on fairness attitudes[17], which shows that the momentary expression of these preferences remains highly contextualised and that our analyses can only offer a scientific abstraction.

### Limitations

Of course, our study is not without limitations. One potential moderating factor in social preferences is social distance[17,88]. Here, we only considered pairs of children that had little or no previous connection. Most decisions, especially in ancestral environment, but also in the modern world, are not about strangers, but about others in the typical interaction group. It is an important question to what extent the fairness attitudes reported here depend on the relationship quality among the interacting children. Furthermore, the data of this study were collected by one single female (cis-gendered) experimenter. We cannot evaluate whether our results would have been different with a male researcher. Another limitation is the use of biological sex as a proxy for a binarized gender construct. We have no evidence that any child was misgendered, so we assume that in our sample, these labels are a close match. However, this approach also ignores diversity on the gender identity spectrum, which should be studied in relation to gender-related fairness preferences as well. Finally, we used smiley stickers as rewards. It is possible that some of our findings can be explained by potential differences in valuation of the stickers. For example, boys might have been less spiteful with girls compared with other boys because they possibly believed that girls would be more interested in stickers than boys. Moreover, the interest in stickers might change with age, which could potentially explain the age group differences, especially regarding advantageous inequity aversion. Finally, it is important to note that our study was not designed to provide a mechanistic explanation for our findings; we first needed to demonstrate that fairness preferences depend on the partner's gender at all. Future endeavours should address these open questions.

### Conclusion

Gender stereotypes permeate today's society. Our study highlights the pervasiveness of gendered differences in social behaviour, even in young children, possibly contributing to cultural gender stereotypes in adult life. However, as our study shows, at least in the field of fairness preferences, gendered differences solidify over an extended period. This observation also leaves room for promoting non-gender-stereotyped fairness attitudes during this critical period.

### Data availability

Raw data is available open access in the OSF repository for this project (https://osf.io/pk6h5/)[89]. The data[74] from the original study by Blake et al.[16] is available via DataDryad.

## Code availability

Data was analysed and edited using Matlab R2016 (MathWorks, Natick, Massachussets, U.S.A.), IBM SPSS Statistics 22 (IBM, New York, U.S.A.), RStudio 2021.09.0, R 4.1.3, GIMP and Inkscape. R packages include: • Effects 4.2-2[70–72] • Tidyverse 1.3.1[90] • lme4 1.129[69] and data I/O packages. All code used for analysis and figure generation is available open access in the OSF repository for this project (https://osf.io/pk6h5/)[89].

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

## Acknowledgements

Lina Oberließen was supported by the German Academic Scholarship Foundation. Marijn van Wingerden was supported by the Volkswagen Stiftung Freigeist fellowship, AZ88216. The work was supported by a grant from the German Research Foundation (Deutsche Forschungsgemeinschaft, DFG) awarded to Tobias Kalenscher (KA 2675/5-3). We would like to thank Katrin Schlauch for her invaluable contribution to this study. The funders had no role in study design, data collection and analysis, decision to publish or preparation of the manuscript

## Author contributions

M.v.W.: performed data analysis, wrote final version and revision of the manuscript. L.O.: conceptualised the study, collected the data, drafted initial

version of manuscript. T.K.: provided funding for the experiment, wrote final version and revision of the manuscript

## Funding

## Competing interests
The authors declare no competing interests.
