## [transparent peer review · Communications Psychology]

Decision letter and referee reports: first round

Dear Dr van Wingerden,

Thank you for your patience during the peer-review process. Your manuscript titled "Egalitarian preferences in young children depend on the sexes of the interacting partners" has now been seen by 3 reviewers, and I include their comments at the end of this message. They find your work of interest but raised some important points. We are interested in the possibility of publishing your study in *Communications Psychology*, but would like to consider your responses to these concerns and assess a revised manuscript before we make a final decision on publication.

We therefore invite you to revise and resubmit your manuscript, along with a point-by-point response to the reviewers. Please highlight all changes in the manuscript text file.

Editorially, we consider it important that the revised manuscript clarifies how sex/gender was determined for your participants and uses consistent language to describe the measured concept. We encourage to review the SAGER guidelines for the use of sex/gender, <https://link.springer.com/article/10.1186/s41073-016-0007-6>. In addition, we request an improved literature review regarding sex/gender differences in economic studies involving children. We encourage you to consider Reviewer 1's suggestions for additional analyses that would broaden the scientific significance and appeal of the work. The description of the findings regarding the Fehr-Schmidt concept should be aligned to the model. Please provide a sensitivity analysis regarding the power to detect age-related effects in your sample.

I am attaching an Editorial Requests Table that details critical reporting requirements for the revised manuscript. Please attend to each item and ensure your manuscript is fully compliant. We are requesting that your manuscript aligns with these requirements as this facilitates the evaluation of your manuscript, reducing delays in re-review and potential future acceptance. If your revised manuscript is not aligned with these requests on major issues, such as those concerning statistics, it may be returned to you for further revisions without re-review. Additional information can be found in our style and formatting guide Communications Psychology formatting guide.

Please use the following link to submit your

- revised manuscript,
- point-by-point response to the referees' comments,
- cover letter (as a separate document),
- the Editorial Policy Checklist (see below),
- the Reporting Summary (see below), and
- the completed Editorial Request Table (attached):

(link Redacted)

Best regards,

Jennifer Bellingtier

Jennifer Bellingtier, PhD
Senior Editor
Communications Psychology

REVIEWER EXPERTISE:

Reviewer #1 Sociocognitive development

Reviewer #2 Sociocognitive development

Reviewer #3 Economic Psychology

REVIEWER REPORTS:

Reviewer #1 (Remarks to the Author):

This ms presents a single, large-sample study investigating gender differences in the development of advantageous and disadvantageous inequity. The authors used a real-world AI/DI paradigm in which people were partnered with peer children that were either same or opposite-genders. Among the strengths of the manuscript is the large sample size, the real world paradigm, and the importance of the research question: many studies on the development of fairness gloss over gender differences, or do not report them at all, and this kind of large-scale study is much needed to partly sort out where these gender differences emerge and where they begin.

I'm enthusiastic about the need for this kind of study, though I have some suggestions and overall reservations about the approach that I'll detail below:

First, it is somewhat difficult for me to reconcile these findings with what has been found before: Given that this study employs a paradigm that is nearly identical to one that has been used before, I think the authors need to start with a review of gender differences in prior work - some of the data from these studies are available online (e.g., Blake, McAuliffe, et al., 2015), and it would be helpful if the authors could at least briefly review what has been done previously, and whether there were gender differences found in paradigms that were nearly identical. As a more ambitious take, the authors could consider conducting a mini meta-analysis on prior work - at the very least, I think we need to reconcile the data presented here with other data out there in order to avoid any one article being the definitive source on gender differences.

To be clear, most of the studies previously done use gender-matched peers, to my knowledge. So, this work would be novel in that it does a crossing of genders as allocators and recipients. Nonetheless, at least 2 of the 4 conditions the authors ran have been reported/run before (boy-boy pairs and girl-girl pairs), and the authors could look at gender differences in development in those.

Secondly, my main reservation is that there are quite a few open interpretations of these data - ranging from early socialization of girls to believe they don't deserve to have more than others, to girls being more attuned to discrepancies between allocators and recipients, to girls being more socially competitive, etc. There are no mechanisms proposed, and none tested. I don't exactly think the authors can tackle all of that without additional data, but I think a satisfying developmental study needs to at least explore potential mechanisms.

Finally, my last concern is that a lot of results hinged on small effects for one condition, that are relatively inconsistent across ages. Maybe I misunderstood something about the figures, but there doesn't seem to be a consistent story with AI developing earlier in girls, as is stated in the abstract ("girls more than boys aimed to reduce advantageous inequality") - 5-6-year-olds don't show this pattern, and girls only show this pattern at 3-4, and not 5-6.

I think one thing that could be helpful is to show the data using age continuously (rather than in brackets) in Figure 2. As a minor concern, Figure 2 is somewhat difficult to make sense of, because there is a lot of information - I think showing age continuously on the x-axis and likelihood of selecting the equal payoff option might be an easier way to go.

Minor concerns:

- I think the authors are technically referring to gender, not sex
- Figure 2: I think "fraction" might be the wrong word here. Maybe "proportion of children"?

Reviewer #2 (Remarks to the Author):

Humans reject advantageous distributions, which is a rare case among other species. Using a resource allocation game, the authors investigated among three to eight-year-old children the difference in egalitarian between boys and girls when interacting with a same-sex or cross-sex peers. They found a sex difference in response to advantageous inequity aversion, an envy bias in girls (that girls are more aversive to advantageous inequity with boys than with girls), and a sex spite gap.

The research topic is of great significance and the results are intriguing. Commendably, the authors collected data across several subgroups of children to address this question.

However, several concerns arise regarding the conclusions drawn and the adequacy of the data and analyses

1. The abstract claims to have 'found sex effects that gradually emerging during childhood',

implying an interaction between age and sex of both allocators and recipients. However, subsequent sentences addressing this conclusion only describes sex effects without explicitly addressing age.

2. The authors seems to use the terms 'sex' and 'gender' interchangeably in some parts of the manuscript. However, 'sex' refers to biological physical differences while 'gender' refers to how people identify which is more related to socialization. As the authors mentioned, they did not have any measure of gender stereotype attitudes, caution should be exercised when interpreting results in terms of 'gender'.

3. While the inclusion of data from children as young as three to four years old is commendable, the small sample size, particularly in this age group, limits the ability to draw robust conclusions regarding group differences.

4. I had a hard time understanding table 3, hence not following how these data support the claim that 'we confirmed our descriptive report above that IA increases with age in young children, and that disadvantageous IA develops earlier than advantageous IA'(lines 380-381).

5. Multiple subgroup tests were conducted, necessitating transparency regarding the correction procedure for multiple comparisons. Exact p-values should be reported before and after correction, along with effect sizes where applicable, rather than simply indicating significance as 'p < .05' or 'p < .01'.

6. The methods section should provide detailed explanations of the statistical tests employed for Figure 2. Additionally, the figure itself should be revised to improve clarity, as the current presentation makes it difficult to discern differences between subgroups.

Minor:

Typos (e.g., Figure 3C) should be checked and corrected.

Reviewer #3 (Remarks to the Author):

Summary:

The paper looks at other-regarding (social) preferences in young children. It uses incentivized (stickers as incentives) experiments to analyze age differences in social preferences and the role of the sex of the interaction partner. Children (aged 3 to 8 years old) decide in simple allocation tasks on the allocation of stickers. The authors find both age differences and differences in how girls and boys take decisions dependent on the sex of their interaction partner. Girls tend to tolerate less when the allocation is in favor of a boy than in favor of a girl. Boys treat other boys with more spite than girls, whereas a similar pattern of behavior cannot be detected for girls.

Assessment:

The paper uses a meaningful experimental design and setup. Most importantly, it empirically addresses a very relevant research question that has – to the best of my knowledge – not been assessed so far in a similar way. This is an important asset of the paper itself, even if one can, as always, discuss some of the design choices and their (potential) consequences for the results. The empirical analysis is mostly sound, and the paper is very well written. Let me, in the following, discuss a few concerns and suggestions that I have with the current version of the paper.

1. One of the main problems that I have is about the inappropriate use of some economic concepts. Sometimes this is okay for a non-economic journal, but I would advise the authors to go through the entire paper (in particular, the results and discussion sections) and look for a potential misinterpretation of the Fehr-Schmidt (FS) concept. In essence, the FS model is a utility model that takes the members' of a reference group (in a pair: the interaction partner's) outcomes, more precisely, the differences in outcomes between the allocator and the recipients, into account, obviously regardless of the sex of the interaction partner. Alpha and beta in the model are individual utility parameters, but not the description of an allocation. Sentences such as “aligning their alpha-levels with that of their recipient” or “girls were unconditionally inequity averse against advantageous IA...” (line 435ff) are misleading. They imply either reciprocal concerns or directly taking into account the utility function of interaction partners. While both motivations or behavioral tendencies might be potentially relevant, the FS model abstracts from them. There are more examples along those lines throughout the text.

2. This is now a matter of taste of the journal and the authors, but I would like to see a bit more of a discussion of the economics literature on two aspects: the sex (gender) of the interaction partner, and social preferences of children.

There is different possible forms of introducing the sex of the interaction partner in a standard experiment; for instance, first names, avatars, the explicit reference to the sex, and non-anonymous interactions have been used. While the last choice is, in general, the most problematic one (because of the loss of control by the experimenter), with young kids it is probably also the most straightforward one. There is a very short discussion on this issue in the paper, but it should be a bit more extensive, discussing briefly the options and the reasons for the choice.

There are lots of studies on social preferences of children that use allocation experiments. I do not expect a full survey, but a bit more background from the economics literature would

benefit the reader. For instance, Sutter et al. (2018) is an example that is also close to the setup both in terms of the experimental design and in terms of the empirical analysis (with somewhat older children).

3. The results section would benefit from a bit more structure. I think that a somewhat more intuitive description of the results based on the descriptive results.

Minor points:

4. How can N be uneven in pairs?

5. Would results change when not excluding 53 observations?

6. Line 135: "For example, to..."

7. Would results change when only the children that act as the first decision makers are taken into account?

8. I think that it is a bit premature to rule out birth order or the number of siblings (and their sexes) as (partly) explanatory. The current study cannot provide evidence for this conclusion.

References (in the report, not in the paper):

Sutter et al. (2018), Social preferences in childhood and adolescence. A large-scale experiment to estimate primary and secondary motivations, *Journal of Economic Behavior & Organization* 146: 16-30.

Author Responses: first round.

We thank the reviewers for their generally positive and very constructive feedback on our paper. We have now revised our manuscript. Below, we have included a point-by-point response to the reviewers' concerns. We hope that we have been able to address these concerns satisfactorily.

Reviewer 1

I'm enthusiastic about the need for this kind of study, though I have some suggestions and overall reservations about the approach that I'll detail below.

Our reply: we really appreciate the reviewer's positive assessment of our manuscript.

1) First, it is somewhat difficult for me to reconcile these findings with what has been found before:

a. Given that this study employs a paradigm that is nearly identical to one that has been used before, I think the authors need to start with a review of gender differences in prior work - some of the data from these studies are available online (e.g., Blake, McAuliffe, et al., 2015), and it would be helpful if the authors could at least briefly review what has been done previously, and whether there were gender differences found in paradigms that were nearly identical. As a more ambitious take, the authors could consider conducting a mini meta-analysis on prior work - at the very least, I think we need to reconcile the data presented here with other data out there in order to avoid any one article being the definitive source on gender differences.

Our reply: we have added the requested literature review. However, in order to keep the introduction short and concise, we have decided to add this review as an extra chapter to the discussion rather than to the introduction. This also helps us to put our own findings into perspective, and discuss them in the context of previous insights. We hope that the reviewer will agree with this decision. If not, we will, of course, be happy to amend the introduction in a further revision.

Regarding the reviewer's suggestion to add a meta-analysis on gender differences in the development of fairness preferences: although we fully agree that such a meta-analysis would be highly informative, we are somewhat hesitant to implement their suggestion. First, such a meta-analysis could only focus on the role of the gender of the acting child since, to the best of our knowledge, the gender of the partner child is not reported in any of the existing publications (unless the dyads were gender-matched, see below). However, as our paper specifically focuses on the role of gender *dyads* in fairness preferences, and not on gender effects per se, such a meta-analysis would be beyond the scope of our study. As we also make our data available with the paper, perhaps a meta-analysis could be conducted in the future.

b. To be clear, most of the studies previously done use gender-matched peers, to my knowledge. So, this work would be novel in that it does a crossing of genders as allocators and recipients. Nonetheless, at least 2 of the 4 conditions the authors ran have been reported/run before (boy-boy pairs and girl-girl pairs), and the authors could look at gender differences in development in those.

Our reply: We have heeded this great suggestion and now include an analysis comparing our results directly to open-access dataset from Blake et al., 2015¹. This dataset also contains gender-matched

dyads (FF and MM), indeed allowing a direct comparison of data and model fits. Though there are some methodological differences, their main results replicate in our sample, and we add some interesting additional re-analyses of their data in direct comparison with ours.

2) Secondly, my main reservation is that there are quite a few open interpretations of these data - ranging from early socialization of girls to believe they don't deserve to have more than others, to girls being more attuned to discrepancies between allocators and recipients, to girls being more socially competitive, etc. There are no mechanisms proposed, and none tested. I don't exactly think the authors can tackle all of that without additional data, but I think a satisfying developmental study needs to at least explore potential mechanisms.

Our reply: we agree with the reviewer that there are several open interpretations of our data. Our study was not designed to provide a mechanistic explanation for our findings, we first needed to demonstrate that fairness preferences depend on the partner's gender at all. However, in our discussion, we do, in fact, speculate on some potential mechanisms behind our findings, supported by our data (a combination of reinforcement learning and social influences, see below), and we additionally rule out several potential mechanistic explanations such as demographics or socio-economic factors. In addition, we have now added a new section to the discussion in which we argue why we consider biological explanations, such as a genetic pre-disposition, highly unlikely (see: "Biological pre-disposition is an unlikely mechanism for dyad-gender-dependent social preferences"). We conclude that the most probable explanation for our results is a combination of reinforcement learning during past interaction experiences and societal influences (expectations on social gender roles). Finally, we have added a sentence to the 'Limitations' chapter admitting that future studies are necessary to find the exact mechanism.

3) Finally, my last concern is that a lot of results hinged on small effects for one condition, that are relatively inconsistent across ages. Maybe I misunderstood something about the figures, but there doesn't seem to be a consistent story with AI developing earlier in girls, as is stated in the abstract ("girls more than boys aimed to reduce advantageous inequality") - 5-6-year-olds don't show this pattern, and girls only show this pattern at 3-4, and not 5-6.

I think one thing that could be helpful is to show the data using age continuously (rather than in brackets) in Figure 2. As a minor concern, Figure 2 is somewhat difficult to make sense of, because there is a lot of information - I think showing age continuously on the x-axis and likelihood of selecting the equal payoff option might be an easier way to go.

Our reply: with the new analysis of Age as a continuous variable for the GLMMs (to be more in line with previous analysis methods), we hope to have addressed the concerns about inconsistent effects across age groups. Briefly put: we find that the gender of the Allocator interacts with InequityType in influencing inequity rejection, so girls are more likely than boys to reject AI, but not DI (see main Table 3 and Figure 4. This effect, however, does not show a further interaction with Age. If we consider the raw choices (Fig. 2), we indeed see that girls on average reject more AI, but boys follow the same pattern provided they are paired with girls and not with another boy (Fig. 2 FF and MF – most starkly visible in the costly AI condition). As to the question whether AI develops *earlier* in girls, if we consider the model fits (Fig. 6) that were extracted for FF vs. MM in the costly AI and costly DI condition, in both models there is a significant main effect of AllocatorGender, but no further significant interaction with Age. This suggests that the inequity rejection 50-50% threshold might be broken earlier (along the Age axis) in FF compared to MM dyads, but our data provides no evidence that the *development* (i.e. the

rate of change) is different between genders. Whether AI “develops”, this we think depends also on the conditions under which it is tested – had we only tested FF and MM dyads in the costly condition, we could have concluded that AI does not develop (i.e. break the 50-50% indifference line) at all in boys in this paradigm.

4) Minor concerns:

a. I think the authors are technically referring to gender, not sex

Our reply: we have replaced the term ‘sex’ with ‘gender’ throughout the entire manuscript, consistent with the Sex and Gender Equity in Research (SAGER) guidelines². In our original manuscript, we preferably used the term ‘sex’ instead of ‘gender’, for example, when talking about the ‘sexes of the interacting partners’, to highlight the fact that we did not poll the participants based on their gender identity, but solely based on their biological sex. In addition, we have made this decision in order to appear more agnostic about the putative role of culture or society in shaping the sex/gender-dependent social preferences. However, since social preferences are almost certainly not exclusively the result of biological determinants, as we also argue in our paper, we realize that the term ‘sex’ may be inconsistent with the SAGER guidelines. We have therefore replaced this term with the term ‘gender’, except in those instances where we clearly refer to the children’s biological attributes. We have added the following explanatory footnote to the first mention of “gender” in the manuscript’s introduction:

“Note that, in line with the Sex and Gender Equity in Research (SAGER) guidelines for the use of the terms *sex* and *gender*², here, we consistently use the term *gender* (limited to the female/male binary terms “girl” and “boy”) instead of *sex* to reflect the fact that social preferences are most likely shaped by a combination of socio-economic, cultural, experiential and genetic factors. Note, though, that we did not poll or categorize the participants based on their socially constructed (non-binary) gender identity, but solely based on their biological (binary) sex. Data are available disaggregated for all combinations of female/male participants³”.

b. Figure 2: I think "fraction" might be the wrong word here. Maybe "proportion of children"?

Our reply: this terminology has been updated throughout the MS and visualisations.

Reviewer 2

The research topic is of great significance and the results are intriguing. Commendably, the authors collected data across several subgroups of children to address this question.

Our reply: we really appreciate the reviewer’s positive assessment of our manuscript.

1) The abstract claims to have ‘found sex effects that gradually emerging during childhood’, implying an interaction between age and sex of both allocators and recipients. However, subsequent sentences addressing this conclusion only describes sex effects without explicitly addressing age.

Our reply: with the additional analyses of Age as a continuous variable, we can say more about gender differences (that we find) and a possible difference in inequity aversion *development* (i.e., rate of change, see answer to Q-Rev 1.3 above), that we do not find in the GLMM approach (e.g., no interaction between gender and Age x InequityType). For our analyses with the Fehr-Schmidt modelling approach, we found that the envy bias (difference in alpha parameter when modelling choice from FF

vs. FM groups) and the spite gap (difference in beta parameter when modelling choices from FM vs. MM groups) is the strongest (or most consistent for modelling) in the oldest age group. The permutations required to test for a between-age-group difference in a parameter contrast by Allocator or Recipient (by resampling raw choices and permutating between age groups) are quite complex. We have thus rephrased our findings to clearly indicated that we only find evidence for the presence of a significant spite gap and envy bias in the oldest Age group.

2) The authors seems to use the terms 'sex' and 'gender' interchangeably in some parts of the manuscript. However, 'sex' refers to biological physical differences while 'gender' refers to how people identify which is more related to socialization. As the authors mentioned, they did not have any measure of gender stereotype attitudes, caution should be exercised when interpreting results in terms of 'gender'.

Our reply: the editor and reviewer 1 also commented on our use of the terms 'sex' and 'gender'. Following our interpretation of the suggestions by the editor and reviewer, we now made our considerations on preferring Gender over Sex transparent and in line with the 'Sex and Gender Equity in Research (SAGER) guidelines for the use of the terms *sex* and *gender*'². We replaced the term 'sex' with the term 'gender' throughout the manuscript, except in those instances where we clearly refer to the children's biological attributes. We did not explicitly poll children on their gender identity and acknowledge in the manuscript that we used biological sex as a (binary) proxy for gender.

3) While the inclusion of data from children as young as three to four years old is commendable, the small sample size, particularly in this age group, limits the ability to draw robust conclusions regarding group differences.

Our reply: We now include an analysis that uses Age as a continuous variable, which should alleviate concerns about statistical power regarding age. In the Fehr-Schmidt models, the significant effects in alpha/beta parameter contrasts generally show up in the oldest age group, where sample sizes are larger. We now provide a power analysis on the sample size required to detect age group differences in alpha and beta (see supplemental Fig. 8), which shows that our design was sufficiently statistically powered to detect the reported differences in AI and DI between Age groups, even at smaller sample sizes.

To show the dependence of our reported effects for differences in alpha and beta parameters between age groups (main Figure 7) on sample size, we extended the bootstrap method to also take into account the number of samples in each group, ranging from a random draw of N=10 (with a minimum of N=3 participants from each group in the permutation analysis) up until the full combined sample size for the two groups under comparison, in steps of N=5 additional participants. We repeated our bootstrap method (now with N=100 bootstraps per sample size point. This power analysis is shown in supplemental Figure 8. As can be seen from the graphs, our Bonferroni-corrected significant effects (significant increase in alpha from 3-4 to 5-6; significant increase in beta from 5-6 to 7-8) were readily detectable with smaller sample sizes. In addition, the trend in the narrowing of confidence intervals suggests that with an even larger sample, the other between-age group comparisons might have reached significance as well.

0) I had a hard time understanding table 3, hence not following how these data support the claim that 'we confirmed our descriptive report above that IA increases with age in young children, and that disadvantageous IA develops earlier than advantageous IA' (lines 380-381).

Our reply: our results on the significant effect of Age on both DI and AI, and the significant interaction between Age and InequityType are now provided in the models with Age as a continuous variable. We do find that, in the Fehr-Schmidt models, permutation analyses show significant differences for subgroups of 34yo and 56yo in the alpha parameter, but significant differences in the beta parameter only show up between subgroups of 56yo and 78yo. We interpret this as evidence that DI develops earlier (i.e. rises more strongly with age) than AI, in line with the results from the GLMMs.

1) Multiple subgroup tests were conducted, necessitating transparency regarding the correction procedure for multiple comparisons. Exact p-values should be reported before and after correction, along with effect sizes where applicable, rather than simply indicating significance as 'p < .05' or 'p < .01'.

Our reply: We did Bonferroni corrections for multiple comparisons whenever this applied. We report all exact p-values before and after correction.

2) The methods section should provide detailed explanations of the statistical tests employed for Figure 2. Additionally, the figure itself should be revised to improve clarity, as the current presentation makes it difficult to discern differences between subgroups.

Our reply: Figure 2 has been replaced with visualisations of predicted effects from the GLMMs including Age as a continuous predictor (see Fig. 3).

3) Minor: Typos (e.g., Figure 3C) should be checked and corrected.

Our reply: we have carefully checked our manuscript for typos

Reviewer 3

The paper uses a meaningful experimental design and setup. Most importantly, it empirically addresses a very relevant research question that has – to the best of my knowledge – not been assessed so far in a similar way. This is an important asset of the paper itself, even if one can, as always, discuss some of the design choices and their (potential) consequences for the results. The empirical analysis is mostly sound, and the paper is very well written.

Our reply: we thank the reviewer for their favorable comments and their generally positive assessment of our submission.

1) One of the main problems that I have is about the inappropriate use of some economic concepts. Sometimes this is okay for a non-economic journal, but I would advise the authors to go through the entire paper (in particular, the results and discussion sections) and look for a potential misinterpretation of the Fehr-Schmidt (FS) concept. In essence, the FS model is a utility model that takes the members' of a reference group (in a pair: the interaction partner's) outcomes, more precisely, the differences in outcomes between the allocator and the recipients, into account, obviously regardless of the sex of the interaction partner. Alpha and beta in the model are individual utility parameters, but not the description of an allocation. Sentences such as "aligning their alpha-

levels with that of their recipient” or “girls were unconditionally inequity averse against advantageous IA...” (line 435ff) are misleading. They imply either reciprocal concerns or directly taking into account the utility function of interaction partners. While both motivations or behavioral tendencies might be potentially relevant, the FS model abstracts from them. There are more examples along those lines throughout the text.

Our reply: we are now more descriptive in our language around reporting alpha and beta parameters, always indicating that these pertain to model parameters for a given subgroup (and in our case, are not parameters estimated for an individual).

2) This is now a matter of taste of the journal and the authors, but I would like to see a bit more of a discussion of the economics literature on two aspects: the sex (gender) of the interaction partner, and social preferences of children.

There is different possible forms of introducing the sex of the interaction partner in a standard experiment; for instance, first names, avatars, the explicit reference to the sex, and non-anonymous interactions have been used. While the last choice is, in general, the most problematic one (because of the loss of control by the experimenter), with young kids it is probably also the most straightforward one. There is a very short discussion on this issue in the paper, but it should be a bit more extensive, discussing briefly the options and the reasons for the choice.

There are lots of studies on social preferences of children that use allocation experiments. I do not expect a full survey, but a bit more background from the economics literature would benefit the reader. For instance, Sutter et al. (2018) is an example that is also close to the setup both in terms of the experimental design and in terms of the empirical analysis (with somewhat older children).

Our reply: we have added a more extensive discussions of our choice of real, non-anonymous children to the Methods (section *Procedure & Apparatus*). We have also added the requested literature review on the development of social preferences in general, and of gender differences in this development in particular, to our Discussion.

3) The results section would benefit from a bit more structure. I think that a somewhat more intuitive description of the results based on the descriptive results.

Our reply: We hope that the new results structure is more intuitive, as it follows previous analyses more closely.

4) Minor point: How can N be uneven in pairs?

Our reply: When children had strong positive/negative connections, the pair was excluded. As some children in these pairs also had comprehension problems, the accounting was a bit unintuitive. The current description has cleared that up and provides more information on the exact number of exclusions. When we report N, this is the number of allocators (necessarily in a dyad, but some children in a dyad were excluded as an allocator on the basis of comprehension)

5) Minor point: Would results change when not excluding 53 observations?

Our reply: We compared the accept/reject proportions per dilemma in the full sample vs. the final sample and found no significant differences. We include this analysis in the manuscript.

6) Minor point: Line 135: “For example, to...”

Our reply: This typo has been corrected.

7) Minor point: Would results change when only the children that act as the first decision makers are taken into account?

Our reply: We have analysed the effect of being a first-mover in a dyad and found no significant effect of adding this factor in the GLMMs

8) Minor point: I think that it is a bit premature to rule out birth order or the number of siblings (and their sexes) as (partly) explanatory. The current study cannot provide evidence for this conclusion.

Our reply: We have tested the effect of having 1 or more siblings on rejection of inequity, by adding this as an additional factor to the baseline model. Adding this factor did not significantly improve the model fit. We do not provide evidence against the possibility that birth order could explain some or all of our results. This limitation is mentioned in the discussion section.

References

1. Blake, P. R. *et al.* The ontogeny of fairness in seven societies. *Nature* **528**, 258–261 (2015).
2. Heidari, S., Babor, T. F., De Castro, P., Tort, S. & Curno, M. Sex and Gender Equity in Research: rationale for the SAGER guidelines and recommended use. (2016) doi:10.1186/s41073-016-0007-6.
3. Eagly, A. H. Reporting sex differences. (1987).

Decision letter and referee reports: second round

Dear Dr van Wingerden,

Your manuscript titled "Egalitarian preferences in young children depend on the genders of the interacting partners" has now been seen by our reviewers, whose comments appear below. In light of their advice I am delighted to say that we are happy, in principle, to publish a suitably revised version in Communications Psychology under the open access CC BY license (Creative Commons Attribution v4.0 International License).

We therefore invite you to revise your paper one last time to address any remaining concerns of our reviewers and a list of editorial requests. At the same time we ask that you edit your manuscript to comply with our format requirements and to maximise the accessibility and therefore the impact of your work.

EDITORIAL REQUESTS:

SUBMISSION INFORMATION:

OPEN ACCESS:

Communications Psychology is a fully open access journal. Articles are made freely accessible on publication under a CC BY license (Creative Commons Attribution 4.0 International License). This license allows maximum dissemination and re-use of open access materials and is preferred by many research funding bodies.

For further information about article processing charges, open access funding, and advice and support from Nature Research, please visit <https://www.nature.com/commpsychol/article-processing-charges>

At acceptance, you will be provided with instructions for completing this CC BY license on behalf of all authors. This grants us the necessary permissions to publish your paper. Additionally, you will be asked to declare that all required third party permissions have been obtained, and to provide billing information in order to pay the article-processing charge (APC).

* **DATA AVAILABILITY:**

Please use the following link to submit the above items:
(link redacted)

Best regards,

Jennifer Bellingtier

Jennifer Bellingtier, PhD
Senior Editor
Communications Psychology

REVIEWERS' EXPERTISE:

Reviewer #2 Sociocognitive development
Reviewer #3 Economic Psychology

REVIEWERS' COMMENTS:

Reviewer #2 (Remarks to the Author):

The authors have addressed all my concerns. Thanks for their efforts.

Reviewer #3 (Remarks to the Author):

The manuscript has improved considerably over the previous version. The authors responded to all my concerns in a convincing way. I particularly like that the paper is now embedded in a better way into the literature and that the discussion is more nuanced.

I have no further suggestions or concerns.